# RAGEN-2: Reasoning Collapse in Agentic RL

**Zihan "Zenus" Wang** [1][†][*]   **Chi Gui** [1][2][†]   **Xing Jin** [3][†]   **Qineng Wang** [1][†]   **Licheng Liu** [4][†]   **Kangrui Wang** [1]
**Shiqi Chen** [5]   **Linjie Li** [6]   **Zhengyuan Yang** [7]   **Pingyue Zhang** [1]   **Yiping Lu** [1]   **Jiajun Wu** [8]   **Li Fei-Fei** [8]
**Lijuan Wang** [7]   **Yejin Choi** [8]   **Manling Li** [1]

## Abstract

RL training of multi-turn LLM agents is unstable, and reasoning quality drives task performance. Entropy, the standard reasoning-stability monitor, only measures within-input diversity and misses whether reasoning depends on the input. We identify **template collapse**: stable entropy alongside input-agnostic boilerplate, invisible to entropy and existing metrics. We diagnose it via a **mutual-information (MI) proxy** that scores cross-input distinguishability online; across tasks, MI correlates with final performance far more strongly than entropy. We then explain collapse via a **signal-to-noise ratio (SNR)** mechanism: low within-input reward variance weakens task gradients, letting input-agnostic regularization dominate and erase cross-input differences. We mitigate this with **SNR-Aware Filtering**, prioritizing high-variance prompts each iteration. Across planning, math reasoning, web navigation, and code execution, the method consistently improves input dependence and task performance.

## 1. Introduction

Training multi-turn LLM agents with reinforcement learning (RL) is inherently challenging (Qi et al., 2025; Zhang et al., 2026; Yu et al., 2025). Researchers therefore monitor reward for **outcome stability** and entropy for **reasoning process stability** (Schulman et al., 2017b; Ouyang et al., 2022; Xu et al., 2025), treating both as stability indicators of RL training.

However, entropy can be an ambiguous signal to understand reasoning quality. When entropy decreases, it may

simply reflect the model becoming more specialized and confident on the task, which is a natural outcome of RL optimization (Yu et al., 2025; Xu et al., 2025). When entropy remains high, reasoning can still drift toward fixed templates that appear diverse within any single input but are effectively the same across inputs (Figure 1). We call this **template collapse**, a failure mode invisible to both metrics. This risk is especially acute in multi-turn settings: sparse rewards cannot distinguish input-driven reasoning from templated reasoning that merely happens to succeed (Wang & Ammanabrolu, 2025; Wang et al., 2025c), and reasoning chains are hard to get directly supervised (Shao et al., 2024; Cui et al., 2025). As a result, template collapse can persist unnoticed during training, making agents unreliable and silently hurting their reasoning abilities.

To understand and mitigate template collapse, this paper addresses two questions. **(Q1) How to diagnose?** (§2) Entropy-based metrics (Wei et al., 2025; Yao et al., 2025; Yun et al., 2025) track within-input variability but miss input dependence across inputs, so they fail to detect template collapse. We propose a mutual information (MI) proxy (Cover & Thomas, 2006) that scores each reasoning chain against all batch inputs to measure input dependence, without external models. **(Q2) Why does it happen?** (§3) We explain through a signal-to-noise ratio (SNR) lens. Task gradients draw signal from reward differences across within-input trajectories. Sampling noise and input-agnostic regularization (KL divergence and entropy regularization (Schulman et al., 2017b; Xu et al., 2025)) dilute this signal. Low SNR lets noise dominate, erasing cross-input reasoning differences.

To address template collapse, based on the SNR view, we introduce **SNR-Aware Filtering**, which uses reward variance as a lightweight SNR proxy to select high-signal prompts each iteration, without additional supervision. Throughout training, the MI proxy monitors input dependence; across experiments, MI correlates with task performance significantly more strongly than entropy, validating it as a diagnostic for template collapse.

Together, they constitute a diagnostic framework for a systematic failure mode in multi-turn agent RL, validated across planning (Schrader, 2018), mathematical reason-

---

[†]Equal contribution. [*]Project Lead. [1]Northwestern [2]UIUC [3]Independent [4]Imperial College London [5]Oxford [6]University of Washington [7]Microsoft [8]Stanford. Correspondence to: Zihan "Zenus" Wang <zihanw@u.northwestern.edu>, Manling Li <manling.li@northwestern.edu>.

*Proceedings of the $43^{rd}$ International Conference on Machine Learning*, Seoul, South Korea. PMLR 306, 2026. Copyright 2026 by the author(s).

ing (Yu et al., 2023; Katz et al., 2025), web navigation, code execution, and tool use, under multiple RL algorithms, model scales, and modalities. SNR-Aware Filtering consistently improves input dependence and task performance, providing direct support for the SNR mechanism.

Our contributions are summarized as follows:

1. **Identifying template collapse.** We find that template collapse occurs when reasoning appears diverse within inputs but becomes input-agnostic across inputs. We propose a mutual information proxy to detect it without external models.

2. **Explaining template collapse via SNR.** We show that low reward variance weakens task gradients while input-agnostic regularization remains constant, erasing input dependence. We provide gradient decomposition evidence across reward-variance buckets.

3. **SNR-Aware Filtering.** We propose filtering prompts by reward variance before each update. We demonstrate that this improves input dependence and performance across tasks, algorithms, scales, and modalities.

## 2. Template Collapse in Multi-turn Agent RL

### 2.1. Setup and Preliminaries

We study closed-loop multi-turn agent reinforcement learning (Wang et al., 2025c), where a policy $\pi_\theta$ is trained by repeatedly rolling out trajectories under the current policy and environment and updating on the collected experience. At each time step $t$, the agent observes $o_t$, generates a response consisting of reasoning tokens $z_t$ and an executable action $a_t$, and receives reward $r_t$, forming a trajectory $\tau = \{(o_t, z_t, a_t, r_t)\}_{t=1}^T$.

We use $X$ to denote the full context available to the model immediately before generating reasoning at turn $t$: this comprises the system prompt, all prior observations $o_{1:t}$, actions $a_{1:t-1}$, and reasoning tokens $z_{1:t-1}$. We use $Z$ to denote the reasoning token sequence the model generates for that turn, excluding action tokens and boundary markers (e.g., </think>).

The standard PPO/GRPO objective contains regularization terms (KL divergence, entropy bonus) that act uniformly across all inputs regardless of their content:

$$\mathcal{L}(\theta) = \mathbb{E}_{x,\tau}\big[A(\tau, x)\big] - \lambda_{\mathrm{KL}} D_{\mathrm{KL}}(\pi_\theta \| \pi_{\mathrm{ref}}) + \lambda_H H(\pi_\theta),$$

where $A(\tau, x)$ is the advantage.

### 2.2. Rethinking Reasoning Collapse from an Information-Theoretic Lens

**Why entropy is insufficient to measure reasoning quality?** Researchers proxy process stability with entropy and outcome stability with reward, treating both as evidence of healthy training. Stable entropy, however, does not guaran-

tee stable reasoning. Reasoning diversity (marginal entropy) $H(Z)$ decomposes via Cover & Thomas (2006):

$$H(Z) = I(X; Z) + H(Z \mid X), \quad (1)$$

where $I(X; Z)$ is input dependence (**mutual information** between input $X$ and reasoning $Z$), and $H(Z \mid X)$ is within-input diversity (**conditional entropy** of reasoning given input). Entropy metrics proxy $H(Z \mid X)$, but neither captures a decline in $I(X; Z)$: the policy can sustain high $H(Z \mid X)$ while $I(X; Z)$ drops to zero, producing diverse but input-agnostic boilerplate. We call this **template collapse**.

**Reasoning regimes with a mutual information view.** Figure 1 illustrates four reasoning states along these two axes: (i) *Diverse Reasoning* (high $H(Z \mid X)$, high $I(X; Z)$): the desired regime where reasoning is both varied within each input and systematically grounded across different inputs; (ii) *Template Collapse* (high $H(Z \mid X)$, low $I(X; Z)$): superficially diverse but input-agnostic—the systematic blind spot of existing stability metrics; (iii) *Compressed Reasoning* (low $H(Z \mid X)$, high $I(X; Z)$): input-faithful but overly deterministic; and (iv) *Low-Entropy Collapse* (low $H(Z \mid X)$, low $I(X; Z)$): fully degenerate with deterministic and input-agnostic outputs. Among these, Template Collapse is uniquely problematic because entropy-based metrics can remain high while input dependence collapses. Empirically, $I(X; Z)$ correlates significantly more strongly with task performance than entropy does (Figure 9).

### 2.3. Mutual Information Proxy Family

**How do we estimate mutual information?** True mutual information $I(X; Z)$ has no closed form for high-dimensional token sequences, so we propose an empirical proxy $\widehat{I}(X; Z)$ based on retrieval. The intuition: mutual information $I(X; Z)$ measures how much knowing the reasoning $Z$ tells us about which input $X$ produced it. When $I(X; Z)$ is high, different inputs yield distinguishable reasoning patterns—the model adapts its reasoning to the specific problem. When $I(X; Z)$ is low, reasoning becomes input-agnostic: observing $Z$ gives little clue about which $X$ it came from. This is the signature of template collapse. If reasoning truly collapses into templates, it should be easy to detect: a reasoning trace $Z$ generated from input $X_i$ will be equally likely under any other input $X_j$.

**Method: In-Batch Cross-Scoring.** Given $P$ prompts and $G$ reasoning samples per prompt from training rollouts, we compute teacher-forced log-likelihoods for every $(Z_{i,k}, X_j)$ pair, forming the scoring matrix $\mathbf{L}_{i,k,j} = \log p_\theta(Z_{i,k} \mid X_j)$. We extract two length-normalized quantities:

$$\begin{aligned} \mathrm{matched}_{i,k} &= \frac{\mathbf{L}_{i,k,i}}{|Z_{i,k}|}, \\ \mathrm{marginal}_{i,k} &= \frac{1}{|Z_{i,k}|} \log \frac{1}{P} \sum_j \exp(\mathbf{L}_{i,k,j}). \end{aligned} \quad (2)$$

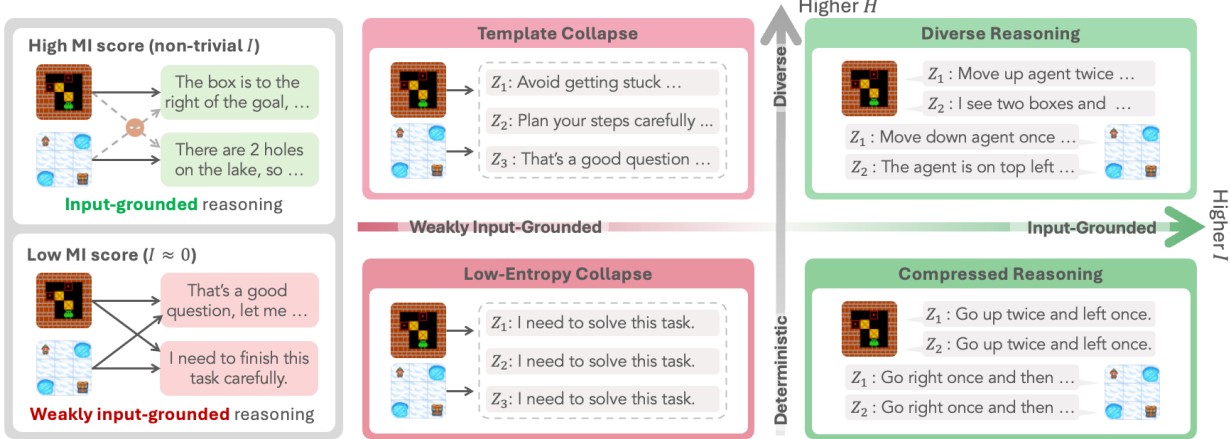

*Figure 1.* Left: input-driven reasoning adapts to the current state; templated reasoning produces nearly identical responses across different inputs. Right: four reasoning regimes characterized along two axes: conditional entropy $H(Z \mid X)$ (within-input diversity) and mutual information $I(X; Z)$ (input dependence). Details in Section 2.

where $\text{matched}_{i,k}$ is the per-token log-likelihood of reasoning $Z_{i,k}$ under its true source input $X_i$, and $\text{marginal}_{i,k}$ approximates the marginal log-likelihood $\log p_\theta(Z_{i,k})$ via a uniform mixture over all prompts in the batch.

**Two Primary Proxies.** We use two complementary proxies derived from Eq. 2:

*(1) Retrieval-Acc (discrete, interpretable):* We define

$$\text{Acc} = \frac{1}{PG} \sum_{i=1}^{P} \sum_{k=1}^{G} \mathbb{I}\Big[i = \arg \max_j \mathbf{L}_{i,k,j}\Big].$$

Under collapse, Acc approaches chance level $1/P$ (1.56% at $P{=}64$), providing an absolute reference.

*(2) MI-ZScore-EMA (continuous, robust):* We estimate input dependence as

$$\widehat{I}(X; Z) = \frac{1}{PG} \sum_{i=1}^{P} \sum_{k=1}^{G} \Big(\text{matched}_{i,k} - \text{marginal}_{i,k}\Big),$$

which increases when reasoning is more compatible with its source input than with the batch mixture. In template-collapse regimes, $\text{matched}_{i,k} \approx \text{marginal}_{i,k}$ for many samples and thus $\widehat{I}(X; Z)$ approaches 0. We apply z-score normalization and exponential moving average (EMA) to stabilize monitoring, yielding MI-ZScore-EMA.

**Proxy Variants and Validation.** Appendix B lists additional proxy variants, varying along three dimensions: (1) turn scope (first-turn only vs. trajectory-uniform sampling); (2) aggregation (discrete retrieval vs. continuous MI estimate); (3) length normalization (per-token vs. per-sequence). For comparison, conditional entropy $H(Z \mid X) = -\frac{1}{PG} \sum_{i,k} \text{matched}_{i,k}$ and marginal entropy $H(Z) = -\frac{1}{PG} \sum_{i,k} \text{marginal}_{i,k}$ are logged, satis-

fying $H(Z) = \widehat{I}(X; Z) + H(Z \mid X)$. We set $\epsilon = 10^{-3}$ and $\alpha = 0.9$ for z-score normalization and EMA, respectively.

Empirically, Retrieval-Acc and MI-ZScore-EMA achieve positive Spearman correlation with final task performance ($+0.39$ for Trajectory MI-ZScore), substantially above entropy metrics, which show negative correlations ($-0.11$ to $-0.14$), confirming entropy is misleading in direction (Figure 9). All proxies reuse $(X_i, Z_{i,k})$ pairs from the training rollout and require no additional model or inference pass; implementation details are in Appendix E.

## 3. The Mechanism of Template Collapse: A Signal-to-Noise Ratio (SNR) View

We have defined template collapse (low $I(X; Z)$, high $H(Z \mid X)$) and introduced an MI proxy to diagnose it. This section explains why RL training produces this failure mode and how to mitigate it. Our core finding: when policy gradient updates are dominated by input-agnostic noise rather than task-discriminative signal—low signal-to-noise ratio (SNR)—reasoning drifts toward templates that appear diverse within each input but ignore cross-input differences.

### 3.1. Observing Signal-Noise Imbalance in RL Gradients

We begin with an empirical observation that motivates the mechanistic analysis. Sorting training prompts by their within-input reward variance $\widehat{\text{Var}}(R \mid X)$ and grouping them into equal-sized buckets, we measure the gradient norms contributed by task objectives / regularization terms (Figure 3). Three patterns are consistent across algorithms:

1. **Task gradient scales with reward variance**: $\|g_{\text{task}}\|$ increases monotonically with bucket RV. High-variance prompts yield strong task-discriminative gradients; low-

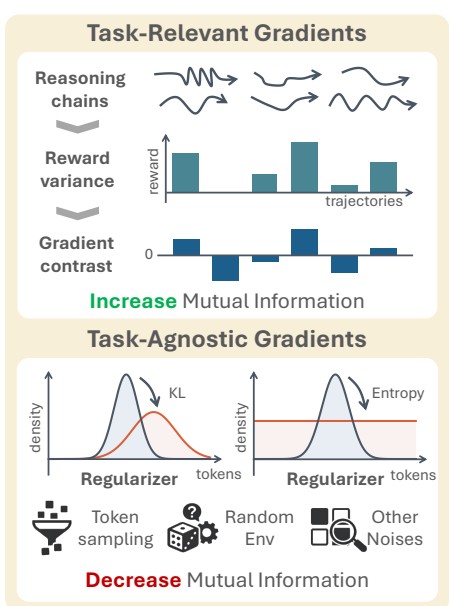

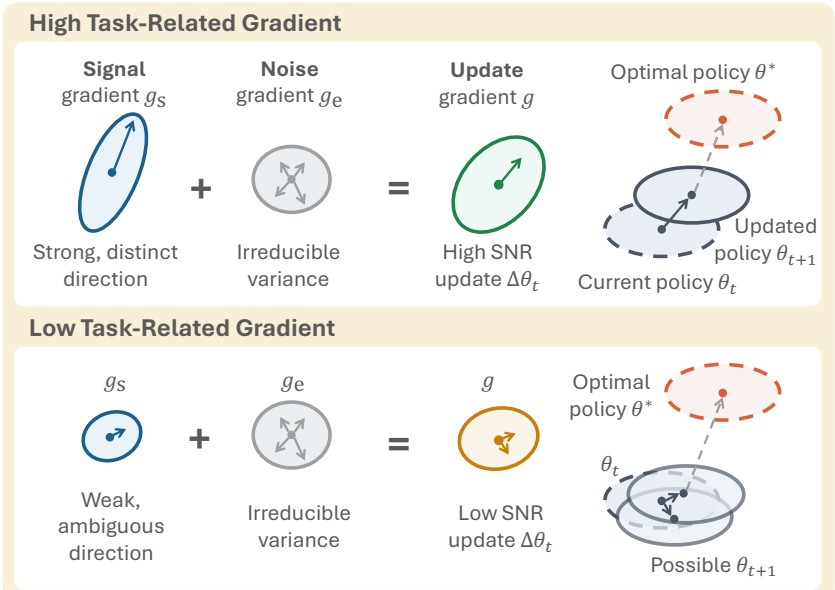

*Figure 2.* Schematic Signal-to-Noise Ratio (SNR) view of RL updates. Left: total gradient decomposes into task gradient (sharpens with higher within-input reward variance) and regularization gradient. Right: high reward variance yields strong task gradient and better convergence (high SNR); low reward variance makes regularization gradient dominate, producing erratic updates and input-agnostic reasoning (low SNR).

variance prompts produce weak gradients.

2. **Regularization gradient is flat**: $\|g_{\text{reg}}\|$ (from KL and entropy terms) remains constant across all buckets, applying uniform contraction to every reasoning chain regardless of its source prompt or reward signal.

3. **Low-RV prompts produce gradient updates dominated by regularization**: In the lowest-variance buckets, task gradients nearly vanish while regularization gradients persist, meaning updates are driven almost entirely by input-agnostic noise.

This gradient imbalance suggests that low reward variance weakens the task-discriminative component of updates, allowing input-agnostic regularization to dominate. When many prompts fall into this regime, the model learns to produce reasoning that satisfies regularization constraints (diverse, fluent) but ignores input-specific requirements—exactly the signature of template collapse.

### 3.2. Formalizing the SNR Mechanism via Gradient Decomposition

The empirical pattern above can be formalized through a *signal-to-noise decomposition of policy gradients*. Low within-input reward variance collapses advantages toward zero, weakening the task gradient. Simultaneously, input-agnostic regularization terms apply uniform contraction to every reasoning chain regardless of its source prompt. When the task gradient is weak, regularization dominates every update and pushes reasoning toward input-agnostic patterns, lowering $I(X; Z)$. This is the gradient-level mechanism behind template collapse (Figure 2; regularizer-dominance

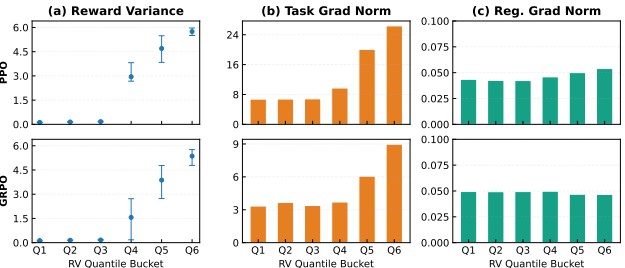

*Figure 3.* Prompts sorted into six reward-variance buckets Q1–Q6: (a) task gradient norm rises monotonically with bucket RV; (b) low-RV task gradients persist but carry almost no useful signal; (c) regularizer gradient norm (KL + entropy) is flat. Supports the SNR mechanism under both algorithms.

analysis in Appendix L).

For input $x$ with $G$ sampled trajectories, the advantage estimate is $A_g = R_g - \bar{R}(x)$ and the task gradient is

$$g_{\text{task}}(x) = \frac{1}{G} \sum_g A_g \, \nabla_\theta \log \pi_\theta(\tau_g \mid x).$$

The Cauchy-Schwarz inequality gives (Appendix I):

$$|g_{\text{task}}(x)| \leq \sqrt{\widehat{\text{Var}}(R \mid X = x) \cdot C}.$$

Low reward variance therefore weakens $g_{\text{task}}$ while leaving $g_{\text{reg}}$ unchanged, driving $I(X; Z) \to 0$. Critically, $H(Z \mid X)$ need not decline: entropy regularization can sustain within-input diversity while input dependence collapses.

We formalize this through a three-noise decomposition of the total gradient:

$$g_{\text{total}} = g_{\text{signal}} + g_{\text{task-noise}} + g_{\text{reg}}.$$

*Table 1.* Three-noise decomposition of the policy update gradient.

| Component | Source | Mitigation |
|---|---|---|
| $g_{\text{signal}}$ | Reward differences across same-prompt trajectories | SNR-Aware Filtering |
| $g_{\text{task-noise}}$ | Sampling and environment stochasticity | Filter high-noise prompts |
| $g_{\text{reg}}$ | Uniform per-chain contraction (KL, entropy) | Tune $\lambda_{\text{KL}}$, $\lambda_{\text{ent}}$ |

*Table 2.* Summary of the features of the environments used.

| Task | Stochastic | Multi-turn | State | Reward |
|---|---|---|---|---|
| Sokoban | ✗ | ✓ | Grid | Dense |
| FrozenLake | ✓ | ✓ | Grid | Binary |
| MetaMathQA | ✗ | ✓ | Text | Dense |
| Countdown | ✗ | ✗ | Text | Binary |
| SearchQA | ✗ | ✓ | Text | Dense |
| WebShop | ✗ | ✓ | Text | Dense |
| DeepCoder | ✗ | ✗ | Text | Dense |

Signal and task noise both vary across prompts, but only the former carries task-discriminative information. Regularization noise acts uniformly at the chain level: every reasoning chain receives the same KL/entropy contraction regardless of its source prompt, making it inherently input-agnostic and the direct suppressive force on cross-input differences (Table 1).

In practice, $g_{\text{task}} = g_{\text{signal}} + g_{\text{task-noise}}$ merges the two prompt-level components, and the SNR is

$$\text{SNR}(x) = \frac{\|g_{\text{signal}}(x)\|}{\|g_{\text{task-noise}}(x)\| + \|g_{\text{reg}}\|}.$$

Low SNR shifts updates toward input-agnostic directions, lowering $I(X; Z)$ even when $H(Z \mid X)$ remains high (Appendix L).

### 3.3. SNR-Aware Filtering: Prioritizing High-Signal Updates

The gradient analysis above identifies the *mechanism behind template collapse*: low reward variance weakens task signal, allowing regularization noise to dominate and push reasoning toward input-agnostic patterns. This suggests a direct mitigation strategy: prioritize prompts with higher within-input reward variance, where advantage estimates carry stronger task-discriminative information and regularization is less likely to dominate the update.

We propose **SNR-Aware Filtering**: at each training iteration, estimate $\widehat{\text{Var}}(R \mid X)$ for each prompt and retain only the top fraction by variance before computing parameter updates (workflow in Figure 4). This concentrates gradient budget on high-SNR prompts and filters out low-variance updates that would be dominated by input-agnostic regularization.

**Reward variance as SNR proxy.** At each iteration, we estimate $\text{Var}(R \mid X)$ at the prompt level by sampling $G$ trajectories for the same prompt $X$ and computing the sample variance of episode returns:

$$\widehat{\text{Var}}(R \mid X) = \frac{1}{G-1} \sum_{g=1}^{G} \left( R_g(X) - \overline{R}(X) \right)^2,$$

$$\overline{R}(X) = \frac{1}{G} \sum_{g=1}^{G} R_g(X).$$

Higher $\widehat{\text{Var}}(R \mid X)$ indicates trajectories can be meaningfully distinguished by reward, strengthening advantage estimates and increasing the likelihood that gradients align with task-relevant directions (Appendix I).

**Top-$p$ filtering by reward variance.** We keep the top fraction of prompts by variance score with keep rate $\rho \in (0, 1]$, analogous to nucleus sampling (Holtzman et al., 2020) but ranking by per-prompt reward variance. Let $V_i = \widehat{\text{Var}}(R \mid X = x_i)$ and let $\sigma$ be a permutation that sorts prompts by descending $V$, so $V_{\sigma(1)} \geq \cdots \geq V_{\sigma(P)}$. With threshold $\tau = \rho \sum_i V_i$, the kept set is $S = \{\sigma(1), \ldots, \sigma(k^*)\}$ where $k^* = \min\{k : \sum_{j=1}^{k} V_{\sigma(j)} \geq \tau\}$, and the filtered objective is $\mathcal{L}_\rho(\theta) = \frac{1}{k^*} \sum_{i \in S} \sum_{j \in \mathcal{B}_i} L_\theta(\xi_j)$. This concentrates updates on high-signal prompts while adapting the kept count to the variance distribution. Other filtering strategies (top-$k$, min-$p$) and implementation details are in Appendix H.

## 4. Experiments

We first establish that template collapse occurs reliably across training configurations (Section 4.2), then evaluate SNR-Aware Filtering as an intervention across tasks, algorithms, model scales, and modalities (Section 4.3; Table 3).

### 4.1. Experimental Testbed

We adopt the RAGEN (Wang et al., 2025c) testbed and evaluate LLM agents on four controllable tasks that stress complementary decision-making regimes: irreversible planning (Sokoban), long-horizon navigation under stochastic transitions (FrozenLake), and symbolic math reasoning (MetaMathQA, Countdown). To further evaluate multi-turn reasoning and decision-making capabilities, we also include SearchQA (Tan et al., 2025), WebShop (Yao et al., 2022), and DeepCoder (Mattern et al., 2025; Li et al., 2023; Jain et al., 2024) (see Appendix C.1 for detailed descriptions).

**Training and evaluation setup.** We train Qwen2.5-3B (Qwen Team, 2024) with the veRL/HybridFlow stack (Sheng et al., 2024), following RAGEN (Wang et al., 2025c) defaults unless otherwise stated. We compare PPO (Schulman et al., 2017b), DAPO (Yu et al., 2025), GRPO (Shao et al., 2024), and Dr. GRPO (Liu et al., 2025) for up to 400 rollout–update iterations. Each iteration collects $K = P \times G = 128$ trajectories per environment, with prompt batch size $P = 8$ and group size $G = 16$ trajecto-

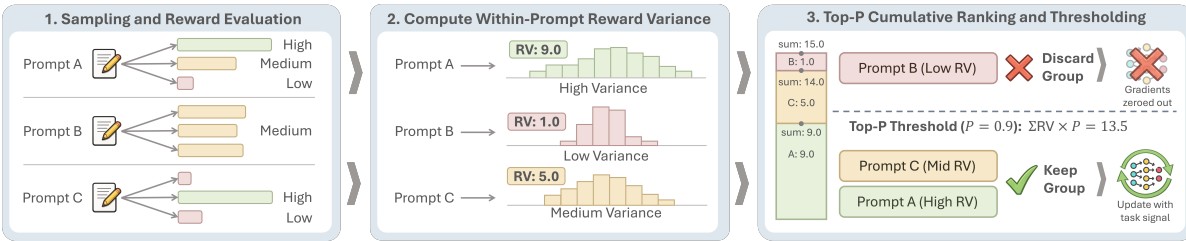

*Figure 4.* SNR-Aware Filtering workflow. At each training iteration: (1) rollout generation collects trajectories; (2) within-prompt reward variance is computed as SNR proxy; (3) prompts are ranked by RV and top-$p$ fraction retained; policy update is performed only on the high-signal subset. The loop prevents updates on noisy rollouts and requires no additional models or rollouts beyond standard RL.

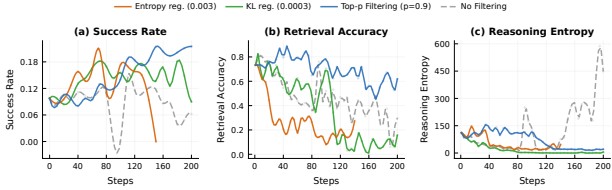

*Figure 5.* Training dynamics under different intervention strategies. (a) Task success rate, (b) MI proxy (retrieval accuracy), and (c) reasoning entropy. Without filtering, MI degrades early while entropy spikes, signaling template collapse. Filtering effectively mitigates the decline in retrieval accuracy, with top-p SNR-Aware filtering best preserving both task performance and reasoning diversity.

ries per prompt. When applying SNR-Aware Filtering with keep rate $\rho$, we reduce the effective minibatch size accordingly and scale the per-step loss by $\rho$, so the optimization step size remains comparable.

### 4.2. Template Collapse as a Consistent Failure Mode

Across all training configurations, RL-trained agents reliably develop reasoning that is fluent but input-agnostic: $I(X; Z)$ declines while $H(Z \mid X)$ remains high, and this drift is invisible to entropy-based monitoring.

**Observing template collapse through MI dynamics.** We track three key metrics during training: task success rate, our MI proxy $\widehat{I}(X; Z)$ (Retrieval-Acc), and conditional entropy $H(Z \mid X)$ (Figure 5). We present dynamics for all MI proxies in Appendix E.1. The trajectory reveals a critical pattern: mutual information declines significantly before task performance degrades, while conditional entropy remains elevated throughout. This divergence is the hallmark of template collapse. Reasoning appears diverse within each input (high $H(Z \mid X)$) but becomes increasingly input-agnostic across inputs (low $I(X; Z)$).

The early decline of $\widehat{I}(X; Z)$ demonstrates that our MI proxy serves as an early warning signal, detecting reasoning degradation that entropy-based metrics miss entirely. This finding motivates using MI as a primary diagnostic alongside task performance, rather than relying solely on entropy for process monitoring.

**Behavioral manifestation of template collapse.** Reasoning length declines monotonically across eight environments

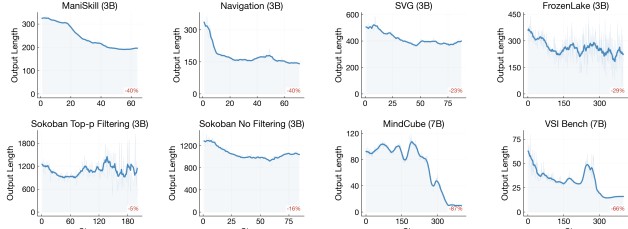

*Figure 6.* Reasoning length decline across eight environments, showing systematic compression as a behavioral signature of template collapse.

spanning spatial (Yin et al., 2025), logic puzzle (Chen et al., 2025), visual (Wang et al., 2025b), and math (Liu et al., 2026) agents (Figure 6), a behavioral signature that complements MI-based diagnostics.

### 4.3. SNR-Aware Filtering Consistently Improves Performance

**Comparing filtering strategies.** Top-$p$ (nucleus-style) filtering consistently outperforms Top-$k$ (fixed-count) and no-filter baselines across four environments (Figure 7). We use Top-$p$ as the default in all subsequent experiments.

Table 3 summarizes our experimental matrix over four tasks, multiple RL algorithms, model scales/types, and input modalities. SNR-Aware Filtering yields two consistent effects. First, it improves peak task success rate in most settings (reported as the $+\Delta$ next to each peak), showing that **prioritizing high-signal updates enhances learning**. Second, the gains span multiple axes, including RL optimizer, model family and scale, and input modality.

Notably, DAPO (Yu et al., 2025) and Dr. GRPO (Liu et al., 2025) are strong baselines targeting stable training and mitigate collapse-like failure modes. Note that DAPO "no-filter" results refer to the original algorithms without our filtering: DAPO also has a filtering step which can be interpreted as a special case of our framework where the selection is fixed (equivalently, a top-$P$ filter with $P \to 1.0$), while our SNR-Aware Filtering provides an explicit, tunable SNR knob via the keep rate $\rho$. This breadth suggests SNR-Aware Filtering serves as a general-purpose SNR control knob and works alongside standard stabilization terms (e.g., KL and entropy

*Table 3.* SNR-Aware Filtering results (%) across algorithms, model scales, types, and modalities. Each cell reports baseline peak with filter delta in parentheses; Qwen2.5-VL-3B includes text (T) and image (V) inputs. Filtering improves average score across all variants.

| Experiment Variants | Sokoban | FrozenLake | MetaMathQA | Countdown | Average |
|---|---|---|---|---|---|
| **Baseline** | | | | | |
| PPO (Schulman et al., 2017b), | | | | | |
| Qwen2.5-3B (Qwen Team, 2024) | 12.9 (+16.0) | 67.0 (+10.9) | 92.6 (+0.6) | 97.9 (+0.0) | 67.6 (+6.9) |
| **Algorithm** | | | | | |
| DAPO (Yu et al., 2025) | 16.2 (+5.1) | 66.8 (+2.1) | 90.8 (+2.8) | 95.7 (+1.6) | 67.4 (+2.9) |
| GRPO (Shao et al., 2024) | 12.1 (+9.0) | 70.9 (-3.0) | 91.2 (+1.2) | 95.7 (+2.2) | 67.5 (+3.7) |
| Dr. GRPO (Liu et al., 2025) | 12.1 (-0.4) | 23.2 (+0.6) | 91.2 (+1.4) | 96.5 (+1.4) | 55.8 (+0.8) |
| **Model Scale (PPO)** | | | | | |
| Qwen2.5-0.5B (Qwen Team, 2024) | 3.3 (+22.9) | 19.5 (+0.0) | 10.0 (-0.2) | 23.0 (-0.7) | 14.0 (+5.5) |
| Qwen2.5-1.5B (Qwen Team, 2024) | 17.0 (+6.2) | 36.5 (+1.6) | 80.3 (+7.0) | 56.6 (+1.6) | 47.6 (+4.1) |
| Qwen2.5-7B (Qwen Team, 2024) | 42.4 (+4.9) | 85.0 (-0.6) | 84.0 (+11.7) | 97.7 (+0.3) | 77.3 (+4.1) |
| **Model Type** | | | | | |
| Qwen2.5-3B-Instruct (Qwen Team, 2024) | 22.5 (+14.2) | 83.6 (+2.3) | 91.2 (+0.4) | 96.3 (-0.6) | 73.4 (+4.1) |
| Llama3.2-3B (Meta Llama, 2024) | 24.4 (+18.8) | 84.6 (-0.2) | 86.1 (+3.7) | 99.2 (-1.2) | 73.6 (+5.3) |
| **Modality (Input Type)** | | | | | |
| Qwen2.5-VL-3B (T) (Bai et al., 2025) | 53.0 (+6.0) | 16.0 (+53.5) | - | - | 34.5 (+29.8) |
| Qwen2.5-VL-3B (V) (Bai et al., 2025) | 65.0 (+12.0) | 19.5 (+59.5) | - | - | 42.3 (+35.8) |

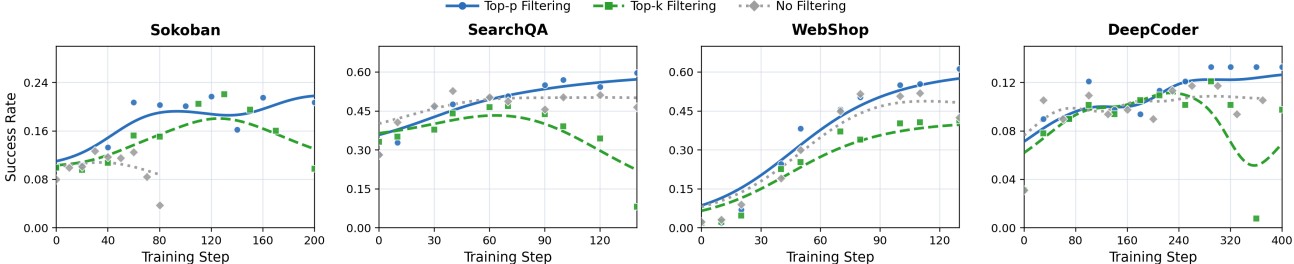

*Figure 7.* Top-$p$ filtering consistently outperforms Top-$k$ and no-filter baselines across four environments.

regularization).

**Compute overhead.** SNR-Aware Filtering needs $G \geq 2$ trajectories per prompt to estimate per-prompt RV; the variance computation itself adds $<0.1\%$ of iteration time, and filtering reduces per-step gradient time by 26–41% at $\rho=0.9$. Full $P \times G$ sweep on Sokoban in Appendix R.1.

## 5. Analysis

### 5.1. MI Diagnoses Collapse Better Than Entropy Across All Interventions

We demonstrate that MI separates high- and low-performance runs across all three intervention families better, and entropy could conflate them. At the same training budget, stronger SNR-Aware Filtering moves runs toward higher MI and better performance; KL and entropy tuning shift entropy without moving MI. We sweep three families of interventions (entropy regularization strength, KL constraint strength, and SNR-Aware Filtering keep rate) and compare their trajectories in both diagnostic spaces at fixed training steps (Figure 8). Entropy- and KL-based stabilizers induce larger changes in $H(Z \mid X)$ than in $\widehat{I}(X;Z)$, and

rarely move the model into the high-$\widehat{I}(X;Z)$ regime with clearly improved performance. In contrast, SNR-Aware Filtering traces a monotone improvement in both $\widehat{I}(X;Z)$ and task success; pushing entropy too high leads to instability and performance collapse, while KL constraint mainly anchors the policy near its reference distribution without boosting input dependence.

We compute Spearman correlation between task success rate and each candidate diagnostic across runs with varying entropy regularization strength, KL constraint strength, and Top-$p$ filtering kept mass (Figure 9). MI-family metrics achieve positive correlations, with Trajectory MI-ZScore reaching +0.39. In contrast, Reasoning Entropy and Conditional Entropy metrics show near-zero or negative correlations (between $-0.11$ and $-0.14$). This confirms that MI predicts performance twice as reliably as entropy does, and entropy actually points in the wrong direction. These results validate MI as a superior training monitor compared to entropy-based diagnostics for multi-turn agent RL.

**Stress-testing the SNR claim.** The SNR account makes concrete causal predictions about how reward variance, environmental noise, and prompt selection should affect $\widehat{I}(X;Z)$

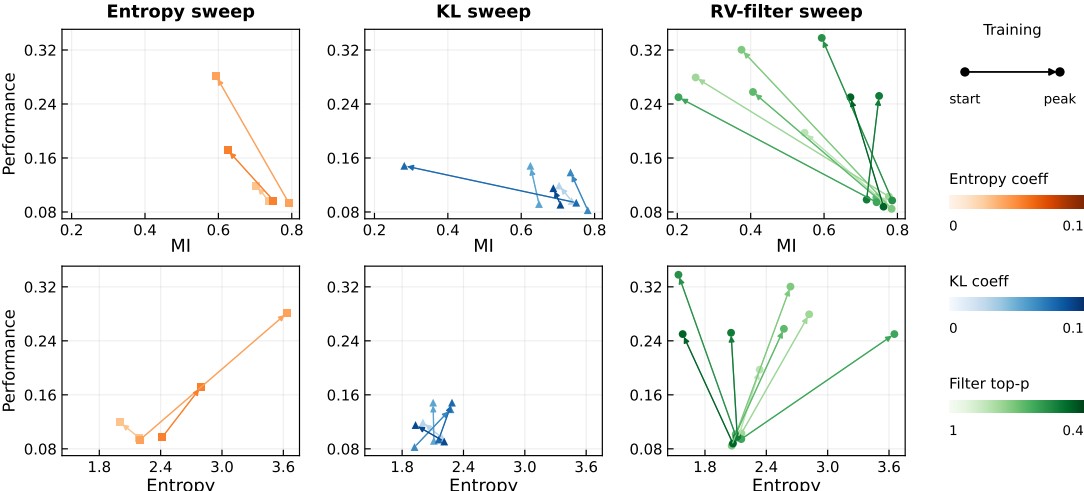

*Figure 8.* Training dynamics under three interventions. For each setting we connect two checkpoints (steps 10/400) into a trajectory (arrows point to later steps); color intensity is weaker to stronger intervention.

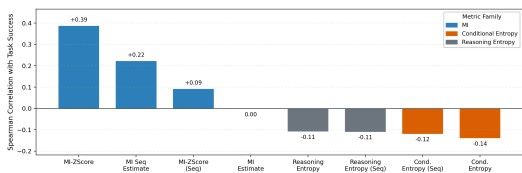

*Figure 9.* Spearman correlations showing MI-family metrics more positively predict performance than entropy metrics.

and task performance. Appendix S verifies these via quartile ablation, controlled noise injection, prompt- vs. trajectory-level filtering, a Std/Mean(RV) applicability diagnostic, the adaptive kept-ratio dynamics, and a comparison against KL/entropy tuning.

## 6. Related Work

**Reasoning collapse and policy degeneracy.** LLM-agent RL reports reasoning collapse (templated rationales) and policy-level degeneracy (behavior concentrating on easy-to-reproduce patterns) (DeepSeek AI, 2025; Wei et al., 2025; Yao et al., 2025; Yun et al., 2025; Feng et al., 2025; Wang & Ammanabrolu, 2025), echoing model collapse in self-training even when average metrics look stable (Gerstgrasser et al., 2024; Shumailov et al., 2024).

**Evaluating reasoning diversity and input dependence.** Most diversity metrics — lexical (Li et al., 2016; Zhu et al., 2018), embedding-based (Pillutla et al., 2021; Tevet & Berant, 2021), uncertainty (Montahaei et al., 2019; Semeniuta et al., 2019) — capture within-input variability without testing whether differences are input-driven (Tevet & Berant, 2021; Yun et al., 2025); recent input-dependence probes include behavioral tests (Gardner et al., 2020; Ribeiro et al., 2020; Zhu et al., 2024) and retrieval-style matching (Morris et al., 2023; Gao et al., 2024; Zhang et al., 2024; Li & Klabjan, 2025). Reasoning faithfulness (Lanham et al., 2023; Turpin et al., 2023; Siegel et al., 2024; Zaman & Srivastava,

2025) asks a different question (whether rationales reflect true decision bases).

**Stabilizing closed-loop Agent RL.** Prior work spans KL/entropy control, clipping, reward shaping, and curricula (Schulman et al., 2017a;b; Haarnoja et al., 2019; Ouyang et al., 2022; Rafailov et al., 2024; Feng et al., 2025; Wang & Ammanabrolu, 2025; Xu et al., 2025; Yao et al., 2025), plus stepwise rewards and self-correction for multi-step agents (Cobbe et al., 2021; Uesato et al., 2022; Madaan et al., 2023; Shinn et al., 2023; Yao et al., 2023; Wei et al., 2025); none target the within-input reward-variance axis we identify.

## 7. Conclusions and Limitations

We find closed-loop multi-turn agent RL can fail silently: reasoning drifts toward fluent but input-agnostic boilerplate while conditional entropy remains stable. We define this as **template collapse**. Built on this, the paper makes three contributions. First, we introduce a mutual information (MI) proxy between inputs and reasoning, which interprets template collapse and tracks task performance better than conditional entropy. To explain why collapse occurs, we propose SNR mechanism in RL and show that low within-input reward variance suppresses task gradients and lets regularization forces dominate, pushing policy outputs toward input-agnostic templates. To address this, we introduce SNR-Aware Filtering to prioritize prompts with reward variance before each parameter update, improving performance on average across tasks, model scales, and modalities and can integrate easily with existing training pipelines.

**Limitations.** The SNR decomposition assumes signal/noise separability; the method needs RV as a reliable signal proxy and degrades in sparse or highly stochastic reward regimes; all experiments are single-agent; aggressive filtering can narrow exploration. Detailed discussion in Appendix T.

## Acknowledgements

We thank Yuxiang Lin for help with RAGEN infrastructure and environments, and Kyunghyun Cho for insightful discussions on the manuscript.

## Impact Statement

This work studies a failure mode in reinforcement learning for LLM agents, proposes a diagnostic metric, and introduces a data-filtering method to mitigate it. The primary applications are improved training stability and reasoning quality for AI agents. We foresee no immediate harmful societal impacts from the contributions themselves. More capable and reliably-reasoning AI agents may carry broader societal implications, but those are not specific to this work.

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

# Appendix Contents

## A. Extended Related Work

***Reasoning collapse and policy degeneracy in closed-loop LM and agent RL training.***    We study a family of degradation phenomena in closed-loop LLM-agent reinforcement learning that has not yet been uniformly defined, but has been repeatedly reported across settings (DeepSeek AI, 2025; Wei et al., 2025). After the model is updated on self-sampled trajectories over time, it may gradually exhibit *reasoning collapse* and *policy-level degeneracy* (DeepSeek AI, 2025; Wei et al., 2025). Here, *reasoning collapse* mainly refers to the rationales, plans, or explanations becoming increasingly templated and less diverse, while their correspondence to the input goal weakens (Wei et al., 2025; Yao et al., 2025; Yun et al., 2025). In contrast, *policy-level degeneracy* refers to behavioral choices concentrating on a small set of easy-to-reproduce action patterns that yield stable scores, with less exploration and less error correction (Feng et al., 2025; Wang & Ammanabrolu, 2025).

This family of phenomena echoes earlier findings in self-training, self-distillation, and iterative fine-tuning on synthetic or model-generated data. When a model repeatedly trains on its own generated distribution, the feedback loop can gradually narrow the effective data distribution, amplify a few high-probability modes, and suppress long-tail behaviors, even when average quality metrics appear stable (Gerstgrasser et al., 2024; Shumailov et al., 2024). In the agent RL setting, closed-loop optimization on on-policy trajectories introduces additional risks, but these risks do not necessarily appear first as an overt failure of the behavioral policy. Instead, a commonly reported pattern is that, even when the agent's external behavior remains effective or yields stable rewards, language-level reasoning expressions can become concentrated earlier. Plans and explanations may converge to a few reusable narrative skeletons, and their alignment with the specific input goal can weaken (Wei et al., 2025; Xu et al., 2025). In other words, reasoning-level degeneration can decouple from policy-level degeneracy, and in some settings it may precede it (Wang & Ammanabrolu, 2025). In multi-turn interaction, related work also describes several visible signatures of this degradation family, such as within-task convergence across repeated rollouts, cross-task templating where different prompts share the same planning or rhetorical skeleton, and late-stage degeneration where later turns become more mechanical or more conservative (Wang et al., 2025a; Xu et al., 2025).

***Evaluating reasoning diversity, input dependence, and reasoning faithfulness.***    Prior work on evaluating *reasoning diversity* often answers how different the outputs are, but less directly answers whether these differences are *systematically driven by the input goal*, which can blur the interpretation of template-like degeneration under closed-loop training (Tevet & Berant, 2021; Yun et al., 2025). Concretely, common metrics range from lexical measures such as *n*-gram statistics and self-BLEU (Li et al., 2016; Zhu et al., 2018), to embedding-based dispersion and distributional distances (Pillutla et al., 2021; Tevet & Berant, 2021), as well as token-level uncertainty proxies and multi-sample coverage or consistency analyses

(Montahaei et al., 2019; Semeniuta et al., 2019). These metrics primarily capture overall randomness or within-input variability, and they are often less sensitive to whether the reasoning distribution changes coherently *across* inputs (Semeniuta et al., 2019; Tevet & Berant, 2021). Other evaluation protocols rely on model scoring or human preference judgments to compare overall response quality, but they are not designed to isolate input-conditioned reasoning differences, and they may conflate prompt-coupled variation with prompt-agnostic surface diversity, especially when outputs converge to shared formats (Kirk et al., 2024; Yun et al., 2025). This leaves a gap for scalable evaluation of whether reasoning is *diagnostic of the input*, which is particularly salient in multi-turn, stochastic environments where a fixed agent policy can produce diverse yet reusable templates (Wang & Ammanabrolu, 2025). Recent work has started to probe input dependence via behavioral tests and local boundary checks (Gardner et al., 2020; Ribeiro et al., 2020), prompt robustness benchmarks (Zhu et al., 2024), and retrieval-style output–input matching or prompt reconstruction signals (Morris et al., 2023; Gao et al., 2024; Zhang et al., 2024; Li & Klabjan, 2025). However, a unified and scalable treatment tailored to closed-loop agent RL remains limited, even as algorithmic work continues to address long-horizon stability and collapse (Feng et al., 2025; Yao et al., 2025).

A closely related line studies *reasoning faithfulness* (explanation faithfulness), which asks whether a rationale reflects the true basis of a decision rather than a plausible post-hoc story (Lanham et al., 2023; Turpin et al., 2023; Siegel et al., 2024; Zaman & Srivastava, 2025). Our question is related but not equivalent: faithfulness emphasizes whether reasoning causally supports a particular decision, while we focus on a different degeneration risk in closed-loop optimization, namely whether reasoning gradually becomes *less sensitive to the input* and drifts toward reusable templates, even when local explanations remain self-consistent (Kirk et al., 2024). This motivates our decomposition of reasoning diversity into within-input variability and cross-input dependence, and our scalable proxy for the latter through an information-theoretic lens.

***Stabilizing multi-turn Agent RL under closed-loop sampling.*** To improve training stability when aligning LLMs and LLM-based agents, prior work has proposed a broad set of algorithmic and system-level techniques. These include KL control or trust-region style constraints, entropy regularization, clipping and normalization in policy-gradient updates, reward shaping and credit assignment, curriculum design, replay or offline–online mixtures, as well as rejection sampling and best-of-$N$ selection (Schulman et al., 2017a;b; 2018; Haarnoja et al., 2019; Stiennon et al., 2022; Ouyang et al., 2022; Rafailov et al., 2024; Sun et al., 2024; Feng et al., 2025; Wang & Ammanabrolu, 2025; Wang et al., 2025a; Xu et al., 2025; Yao et al., 2025). For multi-step agents, researchers have also explored stepwise rewards and intermediate supervision, imitation-to-RL pipelines, and self-correction or reflection signals to support longer-horizon planning and reduce brittle behaviors (Cobbe et al., 2021; Nakano et al., 2022; Uesato et al., 2022; Madaan et al., 2023; Shinn et al., 2023; Wang et al., 2023; Yao et al., 2023; Dou et al., 2025; Wei et al., 2025).

Despite these advances, many stabilization methods are tuned to prevent optimization collapse or to improve overall reward. When the effective learning signal in the closed loop becomes weak or noisy, these methods do not necessarily prevent drift toward prompt-agnostic templates. For example, if most rollouts for the same prompt receive similar rewards regardless of reasoning quality, then the gradient update carries little information about which reasoning path matters (Moskovitz et al., 2023; O'Mahony et al., 2024; Shumailov et al., 2024; Yun et al., 2025). This motivates methods that explicitly manage the balance between task-specific signal and task-agnostic pressure. We adopt a signal-to-noise view of closed-loop updates: we use within-prompt reward variance as a proxy for signal strength, and we filter low-signal samples to maintain an effective SNR, so that exploration and input-conditioned reasoning are less likely to be washed out over long-horizon multi-turn optimization (Romoff et al., 2018; Shao et al., 2024; Tao et al., 2025; Feng et al., 2025; Yao et al., 2025).

## B. Additional MI proxies

We list additional MI proxies in Table 6. All variants are derived from in-batch cross-scoring of reasoning traces against prompts, using matched (per-token log-prob under the true prompt) and marginal (per-token log-prob under the uniform prompt mixture) as base quantities. First-turn variants use only the first agent turn; trajectory variants sample across all turns.

## C. Detailed Experimental Settings

### C.1. Environments and Tasks

We construct a diverse seven-environment testbed to evaluate LLM agents across complementary axes of decision-making complexity, including planning under irreversible dynamics (Sokoban), long-horizon control with non-deterministic tran-

*Table 4.* MI proxy family.

| Type | Proxy | Formula | Notes |
|---|---|---|---|
| Discrete | Retrieval-Acc
Recall@$k$ | $\frac{1}{PG}\sum_{i,k}\mathbf{1}[\arg\max_j \mathbf{L}_{i,k,j}=i]$
$\frac{1}{PG}\sum_{i,k}\mathbf{1}[i\in\text{top-}k_j(\mathbf{L}_{i,k,j})]$ | Chance level $1/P$ under template collapse
$k\in\{2,4,8\}$ |
| Continuous (raw) | MI-Est
MI-Seq-Est | $\frac{1}{PG}\sum_{i,k}(\text{matched}_{i,k}-\text{marginal}_{i,k})$
$\frac{1}{PG}\sum_{i,k}\left(\mathbf{L}_{i,k,i}-\log\frac{1}{P}\sum_j e^{\mathbf{L}_{i,k,j}}\right)$ | Per-token; approaches 0 under collapse
Per-sequence; no length normalization |
| Continuous (z-score) | MI-ZScore
MI-ZScore-EMA | $\frac{1}{PG}\sum_{i,k}\frac{\text{matched}_{i,k}-\text{marginal}_{i,k}}{\sigma_{\text{batch}}+\epsilon}$
$\frac{1}{PG}\sum_{i,k}\frac{\text{matched}_{i,k}-\text{marginal}_{i,k}}{\sigma_{\text{EMA}}+\epsilon}$ | Normalized by current-batch marginal std
$\sigma_{\text{EMA}}^{(t)}=\alpha\,\sigma_{\text{EMA}}^{(t-1)}+(1-\alpha)\,\sigma_{\text{batch}}^{(t)}$ |

sitions (FrozenLake), multi-step symbolic reasoning in mathematics (MetaMathQA, Countdown), multi-turn search and information synthesis (SearchQA), goal-directed web navigation (WebShop), and program synthesis from input-output specifications (DeepCoder). All environments are synthetic and fully controllable, enabling clean analysis of RL learning from scratch without relying on real-world priors.

**Sokoban.** We use the puzzle Sokoban (Schrader, 2018) to study multi-turn agent interaction with irreversible dynamics. The agent must push boxes to designated target locations within a grid-based warehouse. Unlike standard navigation tasks, Sokoban is characterized by irreversibility: boxes can only be pushed, not pulled, meaning a single misstep can create unsolvable dead-ends where boxes become permanently stuck against walls or corners. This requires the agent to reason ahead and plan multi-step sequences before committing to actions. The reward signal encourages both efficiency and accuracy: $+1$ for each box successfully placed on a target, $-1$ for moving a box off a target, $+10$ upon task completion, and $-0.1$ per action as a step penalty. We use procedurally generated puzzles with configurable room dimensions and box counts to ensure diverse training scenarios.

**Frozen Lake.** This environment of FrozenLake (Brockman et al., 2016) combines long-horizon decision-making with deterministic transitions. The agent navigates a grid of frozen tiles to reach a goal while avoiding holes that terminate the episode. We use the 2% random rate variant of Frozen Lake, where each intended action is executed at a 98% probability. Rewards are sparse: only successful goal-reaching trials receive a reward of $+1$, with all other outcomes yielding $0$. The combination of sparse rewards and long-horizon planning makes this environment challenging for credit assignment.

**MetaMathQA.** To evaluate mathematical reasoning capabilities, we include MetaMathQA (Yu et al., 2023), a question-answering task drawn from the MetaMathQA dataset. Each episode presents the agent with a mathematical problem requiring multi-step reasoning—ranging from arithmetic and algebra to word problems and geometry. The agent must produce a final answer, and correctness is determined by exact match with the ground truth. To encourage efficient reasoning, we employ a diminishing reward scheme: correct answers on the first attempt receive full reward (1.0), with rewards halving for each subsequent attempt (0.5, 0.25, . . . ).

**Countdown.** Inspired by the numbers game from the TV show "Countdown" (Katz et al., 2025), this environment tests compositional arithmetic reasoning. The agent is given a target number and a set of source numbers, and must construct an arithmetic expression using each source number at most once to reach the target exactly. For example, given target 24 and numbers $[1, 5, 6, 7]$, a valid solution is $6\times(7-5+1)+6$. Rewards distinguish between format correctness and solution correctness: full reward (1.0) for correct solutions, partial reward (0.1) for expressions that use the correct numbers but yield incorrect results, and zero for malformed expressions.

**DeepCoder.** To evaluate agent capabilities in coding environments, we use DeepCoder, a coding benchmark consisting of competitive programming problems. It was used to train DeepSeek-R1-Distill-Qwen-14B with reinforcement learning. The benchmark draws from three resources: PrimeIntellect (Mattern et al., 2025), TACO(Li et al., 2023), and LiveCodeBench v5 (LCBv5) (Jain et al., 2024). In this environment, agents are required to generate a Python function that solves the given programming problem and passes all hidden and public test cases. During training, rewards are assigned based on the number of test cases successfully passed.

**SearchQA.** To evaluate multi-turn search and question-answering capabilities, we include SearchQA from the RLLM framework (Tan et al., 2025), specifically the Search R1 variant. This environment requires the agent to perform iterative web search and reasoning to answer open-domain questions. The agent must formulate search queries, extract relevant information from retrieved documents, and synthesize answers across multiple interaction turns. Rewards are based on answer correctness and search efficiency, encouraging the agent to balance exploration breadth with reasoning depth.

**WebShop.** We use WebShop (Yao et al., 2022), an interactive e-commerce environment for evaluating goal-directed multi-turn decision-making. The agent is presented with a shopping instruction (e.g., "find a red shirt under \$30") and must navigate a simulated online shopping website by issuing search queries, clicking on products, and selecting appropriate items. The environment features a large action space with realistic product catalogs and requires the agent to perform language understanding, attribute matching, and sequential decision-making. Rewards are assigned based on how well the purchased item matches the specified attributes and constraints.

### C.2. Training and Evaluation Setup

We conduct our main experiments using Qwen2.5-3B and train with four policy-gradient variants—PPO, DAPO, GRPO, and Dr.GRPO—for up to 400 rollout–update iterations on NVIDIA GPUs using the veRL framework, with early stopping enabled as described below. Each iteration collects $K = 128$ trajectories per environment, organized as $P = 8$ prompt groups with $G = 16$ parallel samples per prompt.

**Episode horizons.** To match task structure, the interactive environments (Sokoban, Frozen Lake) use up to 5 interaction turns with 2 actions per turn (10 total actions per trajectory). The single-step reasoning tasks (Countdown, MetaMathQA) use 1 turn with 1 action.

**Optimization.** We use an update batch size of 32 and a per-GPU minibatch size of 4. Policy optimization uses GAE with $(\gamma, \lambda) = (1.0, 1.0)$ and Adam with $(\beta_1, \beta_2) = (0.9, 0.999)$. The actor learning rate is $1 \times 10^{-6}$ and the critic learning rate is $1 \times 10^{-5}$. We apply entropy regularization with coefficient $\beta = 0.001$. For PPO-based methods, we use asymmetric clipping with $\epsilon_{\text{low}} = 0.2$ and $\epsilon_{\text{high}} = 0.28$. We additionally impose a format penalty of $-0.1$ when the agent fails to output a valid structured response (e.g., missing `<think>` or `<answer>` tags).

**Early stopping.** We stop training if either (i) reward-variance collapse is detected—the reward variance drops below 10% of the baseline variance (defined as the mean variance over the first 10 training iterations) for 5 consecutive iterations—or (ii) the validation success rate remains below 1% for 5 consecutive evaluation checkpoints.

**Filtering ablation.** We compare filtered rollouts with `top_p` = 0.9 (keeping the top 90% of trajectory groups ranked by reward variance) against an unfiltered setting.

**Evaluation.** We evaluate on a fixed set of 512 validation prompts per environment and decode with temperature $T = 0.5$ using stochastic sampling. We report success rate as the primary metric across all environments.

## D. Filtering Ablation Results

We conduct our filtering experiments using Qwen2.5-3B model on Sokoban environment. We summarize the filtering ablation results in Table 5. Each row reports the absolute value of each metric, with the change relative to a section-specific baseline shown in parentheses. Within each block, the first row labeled *baseline* defines the reference point, and all deltas are computed relative to that baseline. We report four metrics: **Task Performance**, defined as the maximum validation success rate attained during training; **MI Proxy**, measured as retrieval accuracy at the training step where task performance peaks; **Entropy**, an estimate of reasoning entropy at the same step; and **Collapse**, a binary indicator of whether validation success ever falls below 0.01 during training.

**Sampling Settings.** We first study the interaction between filtering and sampling by varying sampling thresholds while holding the reward-variance (RV) filter fixed. Relative to the `top_p` = 1.0 baseline, reducing `top_p` or `min_p` generally improves task performance while reducing entropy, but with heterogeneous effects on MI retention. In contrast, `top_k` sampling induces a sharper trade-off: MI proxy is often preserved or improved, while gains in task performance are less consistent. These results indicate that filtering behavior is strongly modulated by the sampling regime, even when the underlying filter metric is unchanged.

**Filtering Metrics.** Next, we fix the sampling scheme and vary the filtering criterion. Switching between RV, entropy-based, entropy-variance, and length-based filters leads to substantial differences in both peak task performance and MI proxy. In particular, SNR-Aware Filtering consistently achieves strong task performance while better preserving MI compared to entropy-based alternatives. Entropy- and length-based filters either suppress MI or fail to prevent collapse, suggesting that reward variance provides a more stable and informative signal for selecting useful rollouts.

**Keep Strategy.** Finally, we compare *keep-largest* and *keep-smallest* strategies under the same `top_k` configuration. As

*Table 5.* Ablation results for sampling strategies, filtering metrics, and keep strategies. Values in parentheses denote the change relative to the corresponding baseline in each block. A crossmark in the Stable column indicates training collapse.

| EXPERIMENT SETUP | TASK PERF | MI PROXY | ENTROPY | STABLE |
|---|---|---|---|---|
| **Sampling Strategies** | | | | |
| Top-p = 1.0 (Baseline) | 0.17 | 0.54 | 2.76 | ✗ |
| Top-p = 0.9 | 0.38 (**+0.20**) | 0.84 (**+0.29**) | 1.64 (-1.12) | ✓ |
| Top-p = 0.5 | 0.29 (**+0.12**) | 0.83 (**+0.29**) | 1.88 (-0.88) | ✓ |
| Min-p = 0.05 | 0.42 (**+0.25**) | 0.67 (**+0.13**) | 1.64 (-1.12) | ✓ |
| Min-p = 0.2 | 0.45 (**+0.27**) | 0.36 (-0.18) | 3.01 (**+0.26**) | ✓ |
| Top-k = 0.25 | 0.22 (**+0.05**) | 0.86 (**+0.32**) | 1.28 (-1.48) | ✓ |
| Top-k = 0.5 | 0.44 (**+0.27**) | 0.89 (**+0.35**) | 1.47 (-1.29) | ✓ |
| **Filtering Metrics** | | | | |
| No Filter (Baseline) | 0.17 | 0.54 | 2.76 | ✗ |
| Reward Variance | 0.38 (**+0.20**) | 0.84 (**+0.29**) | 1.64 (-1.12) | ✓ |
| Reward Sum | 0.24 (**+0.07**) | 0.80 (**+0.26**) | 4.18 (**+1.42**) | ✗ |
| Entropy | 0.20 (**+0.02**) | 0.41 (-0.14) | 2.20 (-0.56) | ✗ |
| Entropy Variance | 0.23 (**+0.06**) | 0.70 (**+0.16**) | 2.94 (**+0.18**) | ✗ |
| Length | 0.16 (-0.02) | 0.91 (**+0.36**) | 1.65 (-1.10) | ✗ |
| **Keep Strategies** | | | | |
| Keep Largest (Baseline) | 0.44 | 0.89 | 1.47 | ✓ |
| Keep Smallest | 0.29 (-0.15) | 0.47 (-0.42) | 5.31 (**+3.84**) | ✓ |

expected, retaining high-variance trajectory groups yields substantially higher task performance and MI proxy, while keeping the smallest-variance groups degrades both and markedly increases entropy. This asymmetry supports the hypothesis that high-variance rollouts contain more informative training signal, whereas low-variance rollouts are largely uninformative or noisy.

**Summary.** Overall, the ablation reveals strong interactions between sampling strategy and filtering choice. More aggressive filtering is not universally beneficial, and the choice of filtering metric is critical: reward-variance filtering consistently improves task performance while maintaining information content, whereas entropy-based heuristics are less reliable and more prone to collapse.

# E. Additional Experimental Visualizations

This section presents supplementary visualizations that provide deeper insights into the mechanisms and diagnostics discussed in the main paper. These figures complement the core experimental results with detailed breakdowns of gradient dynamics, diagnostic validity, and reward distribution patterns.

### E.1. MI Proxy Metrics During Training

Figure 10 presents six alternative mutual-information proxy metrics tracked over the course of training, complementing the retrieval accuracy shown in Figure 5 of the main paper. All proxies exhibit a consistent pattern: under the *No Filtering* baseline, MI proxies degrade sharply as training progresses, while the three intervention strategies (entropy regularization, KL regularization, and top-$p$ filtering) maintain stable information retention throughout training.

# F. Notation and basic identities

### F.1. Random variables and distributions

**Definition F.1** (Prompts, trajectories, and rollouts). Let $X$ denote an input prompt and $Z$ a reasoning trajectory. A rollout sample is

$$\xi = (x, z, r),$$

with $x$ the prompt, $z$ the realized trajectory, and $r \in \mathbb{R}$ the scalar reward.

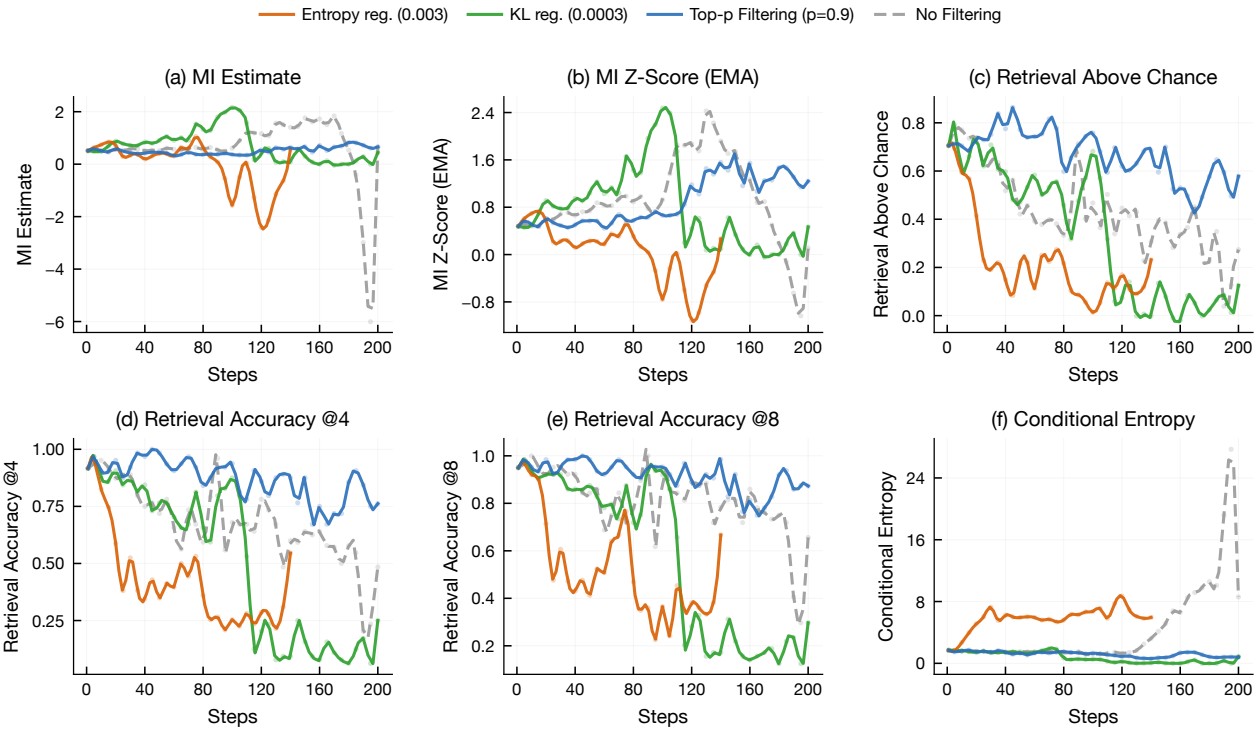

*Figure 10.* Six MI proxy metrics over training steps. (a) MI Estimate, (b) MI Z-Score (EMA), (c) Retrieval Above Chance, (d) Retrieval Accuracy @4, (e) Retrieval Accuracy @8, (f) Conditional Entropy $H(Z|X)$. Consistent with Figure 5, all proxies confirm that without filtering, MI degrades early, signaling reasoning collapse. Filtering effectively preserves information retention across all metrics, with top-$p$ SNR-aware filtering best maintaining reasoning diversity throughout training.

We write $\pi_\theta(z \mid x)$ as the policy and $P(X)$ the prompt distribution. Rollouts are generated by

$$x \sim P(X), \qquad z \sim \pi_\theta(\cdot \mid x), \qquad r = R(z; x),$$

where $R(z; x)$ is the reward function.

**Definition F.2** (Baseline and advantage). Let $b(x)$ be any function of $x$ only. Define the advantage

$$A(z; x) := R(z; x) - b(x).$$

A standard choice is the conditional-mean baseline $b(x) := \mathbb{E}[R(Z; x) \mid X = x]$. Then the advantage is zero-mean within each prompt:

$$\mathbb{E}[A(Z; x) \mid X = x] = \mathbb{E}[R(Z; x) \mid X = x] - b(x) = 0.$$

**Definition F.3** (Score function). Define the score function

$$s(z; x) := \nabla_\theta \log \pi_\theta(z \mid x).$$

It satisfies the normalization identity

$$\mathbb{E}_{z \sim \pi_\theta(\cdot \mid x)}[\, s(z; x) \,] = \nabla_\theta \int \pi_\theta(z \mid x) \, dz = 0.$$

**Definition F.4** (Within-prompt reward variance). We quantify within-prompt variation of observed rewards across rollouts by

$$\mathrm{RV}(x) := \mathrm{Var}(R(Z; x) \mid X = x), \qquad Z \sim \pi_\theta(\cdot \mid x).$$

Low $\mathrm{RV}(x)$ implies rewards are nearly constant within the prompt, so rollouts are weakly distinguishable by the reward signal. High $\mathrm{RV}(x)$ indicates large within-prompt variation of observed rewards which may arise from trajectory-dependent signal or evaluation noise.

## F.2. Entropy and mutual information

**Definition F.5** (Conditional entropy). The within-input variability of reasoning is measured by
$$H(Z \mid X) := \mathbb{E}_{x \sim P(X)}\big[H(Z \mid X = x)\big] = -\mathbb{E}_{x \sim P(X),\, z \sim \pi_\theta(\cdot \mid x)}\big[\log \pi_\theta(z \mid x)\big].$$
The cross-input dependence of reasoning is measured by
$$I(X; Z) := \mathbb{E}_{x \sim P(X),\, z \sim \pi_\theta(\cdot \mid x)}\left[\log \frac{\pi_\theta(z \mid x)}{p_\theta(z)}\right], \qquad p_\theta(z) := \mathbb{E}_{x \sim P(X)}\big[\pi_\theta(z \mid x)\big].$$
Equivalently, $I(X; Z) = \mathbb{E}_{x \sim P(X)}\big[\mathrm{KL}(\pi_\theta(\cdot \mid x) \,\|\, p_\theta)\big]$.

**Decomposition identity (Shannon quantities).** For the true distribution induced by $\pi_\theta$, the Shannon identity
$$H(Z) \;=\; H(Z \mid X) \;+\; I(X; Z), \tag{3}$$
serves only as conceptual equation: it specifies the two components we aim to track (within-prompt variability and cross-prompt dependence). In practice we replace these Shannon quantities by scorer-defined proxies, e.g.,
$$\widehat{\mathcal{D}}_q := \widehat{\mathrm{NLL}}_q(Z \mid X) + \widehat{I}_q(X; Z),$$
which is in log-likelihood units under $q$ and does not in general satisfy the Shannon identity unless $q$ matches the evaluated distribution.

**Interpretation for reasoning diversity.** In our setting, $Z$ is a proxy for a reasoning process (e.g., a chain-of-thought trajectory). A relative decrease in $H(Z \mid X)$ indicates within-prompt concentration of $\pi_\theta(\cdot \mid x)$ (entropy collapse). A relative decrease in $I(X; Z)$ indicates weakened input dependence, i.e., trajectories become less diagnostic of $x$. In our analysis, this can occur when reward-driven updates are weak (e.g., low $\mathrm{RV}(x)$) and the total update is dominated by *reward-agnostic* components (e.g., KL/entropy regularizers). We therefore track these two axes separately; in experiments we use scorer-defined proxies for $H(Z \mid X)$ and $I(X; Z)$.

# G. Scorer-based Proxies for Reasoning Diversity

## G.1. Setup and notation

We define scorer-based proxies using a fixed collection of prompts and multiple rollouts per prompt. Throughout this appendix, the scorer $q$ is fixed and used for evaluation.

**Definition G.1** (Prompt groups). Using the notation from Definition F.1, sample $P$ prompts $\{x_i\}_{i=1}^P \sim P(X)$. For each prompt $x_i$, sample $G$ trajectories
$$z_{i,k} \sim \pi_\theta(\cdot \mid x_i), \qquad k = 1, \ldots, G.$$
We refer to the set $\{z_{i,k}\}_{k=1}^G$ as a *prompt group*.

**Definition G.2** (Teacher-forced scorer and matched-pair score). Let $q$ be a fixed language model used to score how compatible a trajectory $z$ is with a prompt $x$. Define the matched-pair score
$$\ell_i(z) \;:=\; \log q(z \mid x_i).$$
All proxies in this appendix are built from $\ell_i(z)$ and therefore are measured in log-likelihood units under $q$.

**Definition G.3** (Mixture score across prompts). We evaluate each trajectory $z$ under all prompts $\{x_j\}_{j=1}^P$ and define the mixture score
$$\ell_{\mathrm{mix}}(z) \;:=\; \log\left(\frac{1}{P}\sum_{j=1}^P \exp(\ell_j(z))\right) = \log\left(\frac{1}{P}\sum_{j=1}^P q(z \mid x_j)\right).$$
This is the log-likelihood of $z$ under the uniform mixture over prompts induced by $q$. Equivalently, $\ell_{\mathrm{mix}}(z) = \log\big(\frac{1}{P}\sum_{j=1}^P q(z \mid x_j)\big)$ is the log-probability of $z$ under the empirical prompt mixture.

The quantities defined above depend on the sampled prompt set $\{x_i\}_{i=1}^P$ and on the fixed scorer $q$. They are proxies for within-prompt variability and input dependence of trajectories, and should not be interpreted as exact Shannon entropies or mutual information unless $q$ matches the evaluated conditional distribution.

# H. Formal Definition of the Filtering Operator

**Definition H.1** (Filtering operator). Let $\mathcal{B}$ be a minibatch of samples. A *filtering operator* is specified by:

**(i) Grouping key.** A grouping function $g : \mathcal{B} \to \mathcal{G}$ that assigns each sample $\xi \in \mathcal{B}$ a group label
$$u = g(\xi).$$
For $u \in \mathcal{G}$, define the induced group subset
$$\mathcal{B}_u := \{\xi \in \mathcal{B} : g(\xi) = u\}.$$

**(ii) Group statistic.** A statistic $\phi : \in^{\mathcal{B}} \to \mathbb{R}$ that depends only on the samples in the group, and we write $\phi(\mathcal{B}_u)$ for the value computed from $\mathcal{B}_u$.

**(iii) Selection rule (mask).** Given a threshold $\tau \in \mathbb{R}$, the binary mask is
$$m(u) := \mathbf{1}\{\phi(\mathcal{B}_u) \geq \tau\}.$$

**(iv) Filtered objective.** For a per-sample RL loss $L_\theta(\xi)$, the filtered objective is
$$\mathcal{L}_{\text{filt}}(\theta) = \frac{1}{|\mathcal{B}|} \sum_{\xi \in \mathcal{B}} m\big(g(\xi)\big) L_\theta(\xi).$$

**Remark (post-sampling).** Filtering is applied after sampling and only masks gradients; it does not change the rollout distribution.

**Remark (normalization).** In practice one may normalize by the number of kept samples or kept groups (instead of $|\mathcal{B}|$), which rescales the gradient but does not change which samples contribute nonzero gradients.

## H.1. Filtering Strategy Variants

We compare multiple filtering strategies for selecting high-signal prompt groups. All variants share the same grouping structure (prompts with $G$ rollouts each) and statistic (reward variance $\widehat{\text{Var}}(R \mid X = x_i)$ for group $i$), but differ in the selection rule.

**Top-p (nucleus-style) filtering.** The main method used in this paper. Given keep rate $\rho \in (0, 1]$, rank prompts by descending reward variance and select the smallest prefix whose cumulative variance mass reaches $\rho \sum_i \widehat{\text{Var}}(R \mid X = x_i)$. Formally, let $\sigma$ be the permutation such that $\widehat{\text{Var}}(R \mid X = x_{\sigma(1)}) \geq \cdots \geq \widehat{\text{Var}}(R \mid X = x_{\sigma(N)})$, and define

$$k^* = \min \left\{ k : \sum_{j=1}^{k} \widehat{\text{Var}}(R \mid X = x_{\sigma(j)}) \geq \rho \sum_{i=1}^{N} \widehat{\text{Var}}(R \mid X = x_i) \right\}.$$

The kept set is $S = \{\sigma(1), \ldots, \sigma(k^*)\}$. This adaptive selection concentrates updates on high-variance prompts while automatically adjusting the kept count based on the variance distribution. When the batch contains many near-zero-variance prompts, top-p can reject the entire batch if the threshold cannot be reached, providing a natural safeguard against degenerate updates.

**Top-k (proportional) filtering.** An alternative fixed-proportion baseline. Given $\rho \in (0, 1]$, compute $k = \lfloor \rho N \rfloor$ and select the top $k$ prompts by reward variance:
$$S = \{\sigma(1), \ldots, \sigma(k)\}.$$
Unlike top-p, top-k always retains exactly $k$ groups regardless of the variance distribution. This can be less adaptive: when most prompts have near-zero variance, top-k still keeps the highest-variance subset even if all retained prompts carry weak signal.

**Min-p (threshold) filtering.** Inspired by min-p sampling, this strategy keeps all prompts whose variance exceeds a fraction of the maximum variance. Given threshold parameter $p \in (0, 1]$, define

$$\tau = p \cdot \max_i \widehat{\text{Var}}(R \mid X = x_i),$$

and keep all groups above the threshold:

$$S = \left\{ i : \widehat{\text{Var}}(R \mid X = x_i) \geq \tau \right\}.$$

This directly enforces a minimum quality bar: only prompts within a factor of $p$ of the best prompt are retained. The kept count varies with the variance distribution, making this method highly adaptive but potentially unstable when the maximum variance fluctuates.

**Reverse top-p (low-variance) filtering.** A diagnostic baseline that intentionally selects low-variance prompts. Rank prompts by *ascending* reward variance and select the smallest prefix whose cumulative variance mass reaches $\rho \sum_i \widehat{\text{Var}}(R \mid X = x_i)$. This inverted strategy is used in ablation studies to confirm that high variance is essential for effective updates: training on low-variance prompts should degrade both MI and task performance, validating the SNR hypothesis.

**Implementation notes.** All strategies can be configured to exclude zero-variance groups (setting `include_zero=False`) before selection, which removes prompts where all rollouts received identical rewards. For top-p, we use a small epsilon $\varepsilon = 0.01$ to ensure numerical stability when checking whether the cumulative threshold is reached. Additional implementation details and hyperparameter sensitivity are in the codebase.

# I. RV Controls Task-Signal Magnitude and SNR

## I.1. Setup

We use the policy/score/baseline/advantage notation from Appendix F.

In particular, for a fixed prompt $x$ we write $z \sim \pi_\theta(\cdot \mid x)$, $s(z; x) = \nabla_\theta \log \pi_\theta(z \mid x)$, $A(z; x) = R(z; x) - b(x)$ with $b(x) = \mathbb{E}[R \mid X = x]$, and $\text{RV}(x) = \text{Var}(R \mid X = x) = \mathbb{E}[A^2 \mid X = x]$.

## I.2. Assumption

**Assumption I.1** (Reward decomposition)**.** The observed reward admits a decomposition

$$R(z; x) = \mu(x, z) + \varepsilon, \qquad \mu(x, z) := \mathbb{E}[R(z; x) \mid x, z],$$

where $\mu(x, z)$ is the trajectory-dependent mean reward and $\varepsilon$ is a zero-mean noise term satisfying

$$\mathbb{E}[\varepsilon \mid x, z] = 0, \qquad \text{Var}(\varepsilon \mid x, z) = \sigma^2(x) \geq 0.$$

Moreover, the score $s(z; x) = \nabla_\theta \log \pi_\theta(z \mid x)$ is a deterministic (measurable) function of $(x, z)$.

## I.3. Task-gradient magnitude is RV-controlled

The next result shows that the task-gradient norm for a given prompt is at most proportional to the square root of its within-prompt reward variance $\text{RV}(x)$. In particular, when $\text{RV}(x)$ is small, the task gradient is provably weak.

**Theorem I.2** (Task gradient magnitude is RV-controlled)**.** *Assume the baseline is the conditional mean* $b(x) = \mathbb{E}[R \mid X = x]$, *and* $g_{\text{task}}(x) := \mathbb{E}[A(z; x) \, s(z; x) \mid X = x]$. *Then*

$$\|g_{\text{task}}(x)\| \leq \sqrt{\text{RV}(x)} \sqrt{\mathbb{E}[\|s(z; x)\|^2 \mid X = x]}.$$

*Proof.* Fix a prompt $x$ and take randomness over $z \sim \pi_\theta(\cdot \mid x)$. For brevity write $A := A(z; x)$ and $s := s(z; x)$. Then

$$g_{\text{task}}(x) = \mathbb{E}[A \, s \mid X = x].$$

For any unit vector $u \in \mathbb{R}^d$ with $\|u\| = 1$,

$$\left| \langle u, g_{\text{task}}(x) \rangle \right| = \left| \mathbb{E}[A \langle u, s \rangle \mid X = x] \right| \leq \sqrt{\mathbb{E}[A^2 \mid X = x]} \sqrt{\mathbb{E}[\langle u, s \rangle^2 \mid X = x]},$$

where the inequality is Cauchy-Schwarz. Moreover, $\langle u, s \rangle^2 \leq \|u\|^2 \|s\|^2 = \|s\|^2$, hence

$$\left| \langle u, g_{\text{task}}(x) \rangle \right| \leq \sqrt{\mathbb{E}[A^2 \mid X = x]} \sqrt{\mathbb{E}[\|s\|^2 \mid X = x]}.$$

Taking the supremum over all unit vectors $u$ yields

$$\|g_{\text{task}}(x)\| \leq \sqrt{\mathbb{E}[A^2 \mid X = x]} \sqrt{\mathbb{E}[\|s\|^2 \mid X = x]}.$$

Finally, with $b(x) = \mathbb{E}[R \mid X = x]$ we have $\mathbb{E}[A \mid X = x] = 0$ and thus

$$\mathbb{E}[A^2 \mid X = x] = \text{Var}(R \mid X = x) = \text{RV}(x).$$

Substituting completes the proof. $\qquad\square$

## I.4. SNR is upper bounded by RV and reward noise

The following theorem shows that the signal-to-noise ratio of the $G$-sample Monte Carlo gradient estimator is upper-bounded by $\sqrt{G} \cdot \sqrt{\text{RV}(x)}/\sigma(x)$. When reward variance is low relative to reward noise, the estimator is dominated by noise.

**Theorem I.3** (SNR upper bound by RV and noise). *Let $\widehat{g}_{\text{task}}(x)$ be the $G$-sample Monte Carlo estimator*

$$\widehat{g}_{\text{task}}(x) := \frac{1}{G}\sum_{k=1}^{G} A_k\, s_k, \qquad A_k := A(z_k; x),\ s_k := s(z_k; x),$$

*with $z_1, \ldots, z_G \overset{i.i.d.}{\sim} \pi_\theta(\cdot \mid x)$. Define*

$$\text{SNR}(x) := \frac{\|g_{\text{task}}(x)\|}{\sqrt{\mathbb{E}\big[\|\widehat{g}_{\text{task}}(x) - g_{\text{task}}(x)\|^2 \mid X = x\big]}}.$$

*Under Assumption I.1 and with baseline $b(x) = \mathbb{E}[R \mid X = x]$,*

$$\text{SNR}(x) \ \leq\ \sqrt{G} \cdot \frac{\sqrt{\text{RV}(x)}}{\sigma(x)}.$$

*If $\sigma(x) = 0$, the bound is vacuous.*

*Proof.* Fix a prompt $x$. Let $z_1, \ldots, z_G \overset{i.i.d.}{\sim} \pi_\theta(\cdot \mid x)$ and write

$$\widehat{g} \ = \ \frac{1}{G}\sum_{k=1}^{G} A_k s_k, \qquad g \ = \ \mathbb{E}[As \mid x],$$

where $(A_k, s_k) = (A(z_k; x), s(z_k; x))$ and $(A, s) = (A(z; x), s(z; x))$ for $z \sim \pi_\theta(\cdot \mid x)$.

Let $Y_k := A_k s_k$. Then $\widehat{g} = \frac{1}{G}\sum_{k=1}^{G} Y_k$ and $g = \mathbb{E}[Y_1 \mid x]$, hence

$$\widehat{g} - g = \frac{1}{G}\sum_{k=1}^{G}(Y_k - g).$$

Using i.i.d. conditional on $x$,

$$\mathbb{E}\big[\|\widehat{g} - g\|^2 \mid x\big] = \frac{1}{G^2}\mathbb{E}\left[\left\|\sum_{k=1}^{G}(Y_k - g)\right\|^2 \Bigm| x\right]$$

$$= \frac{1}{G^2}\sum_{k=1}^{G}\mathbb{E}\big[\|Y_k - g\|^2 \mid x\big] + \frac{1}{G^2}\sum_{k \neq \ell}\mathbb{E}[\langle Y_k - g,\ Y_\ell - g\rangle \mid x]$$

$$= \frac{1}{G^2}\sum_{k=1}^{G}\mathbb{E}\big[\|Y_k - g\|^2 \mid x\big]$$

$$= \frac{1}{G}\,\mathbb{E}\big[\|As - g\|^2 \mid x\big].$$

Under Assumption I.1 and with baseline $b(x) = \mathbb{E}[R \mid X = x]$, write $R = \mu + \varepsilon$ with $\mu(x, z) = \mathbb{E}[R \mid x, z]$. Since $b(x) = \mathbb{E}[R \mid x] = \mathbb{E}[\mu \mid x]$,

$$A \ = \ R - b(x) \ = \ (\mu - \mathbb{E}[\mu \mid x]) + \varepsilon \ =:\ A_\mu + \varepsilon.$$

Using $A = A_\mu + \varepsilon$,

$$As - g = (A_\mu s - g) + \varepsilon s,$$

so
$$\|As - g\|^2 = \|A_\mu s - g\|^2 + \|\varepsilon s\|^2 + 2\langle A_\mu s - g, \ \varepsilon s\rangle.$$

Moreover,
$$\mathbb{E}[\langle A_\mu s - g, \ \varepsilon s\rangle \mid x] = \mathbb{E}\Big[\mathbb{E}[\langle A_\mu s - g, \ \varepsilon s\rangle \mid x, z] \ \Big| \ x\Big] = \mathbb{E}\Big[\langle A_\mu s - g, \ s\rangle \, \mathbb{E}[\varepsilon \mid x, z] \ \Big| \ x\Big] = 0,$$

hence
$$\mathbb{E}\big[\|As - g\|^2 \mid x\big] \ \geq \ \mathbb{E}\big[\|\varepsilon s\|^2 \mid x\big].$$

Combining with the variance decomposition above,
$$\mathbb{E}\big[\|\widehat{g} - g\|^2 \mid x\big] \ \geq \ \frac{1}{G}\,\mathbb{E}\big[\|\varepsilon s\|^2 \mid x\big].$$

Since $\|\varepsilon s\|^2 = \varepsilon^2\|s\|^2$ and $s$ is measurable given $(x, z)$,
$$\mathbb{E}\big[\|\varepsilon s\|^2 \mid x\big] = \mathbb{E}\Big[\mathbb{E}\big[\varepsilon^2\|s\|^2 \mid x, z\big] \ \Big| \ x\Big]$$
$$= \mathbb{E}\Big[\|s\|^2\,\mathbb{E}\big[\varepsilon^2 \mid x, z\big] \ \Big| \ x\Big]$$
$$= \mathbb{E}\Big[\|s\|^2\,\sigma^2(x) \ \Big| \ x\Big] = \sigma^2(x)\,\mathbb{E}\big[\|s\|^2 \mid x\big],$$

where $\mathbb{E}[\varepsilon^2 \mid x, z] = \mathrm{Var}(\varepsilon \mid x, z) = \sigma^2(x)$ by Assumption I.1. Therefore
$$\mathbb{E}\big[\|\widehat{g} - g\|^2 \mid x\big] \ \geq \ \frac{1}{G}\,\sigma^2(x)\,\mathbb{E}\big[\|s\|^2 \mid x\big].$$

By Theorem I.2,
$$\|g\| = \|\mathbb{E}[As \mid x]\| \leq \sqrt{\mathrm{RV}(x)}\,\sqrt{\mathbb{E}[\|s\|^2 \mid x]}.$$

Thus, with $\mathrm{SNR}(x) := \frac{\|g\|}{\sqrt{\mathbb{E}[\|\widehat{g}-g\|^2 \mid x]}}$,
$$\mathrm{SNR}(x) \leq \frac{\sqrt{\mathrm{RV}(x)}\sqrt{\mathbb{E}[\|s\|^2 \mid x]}}{\sqrt{\frac{1}{G}\sigma^2(x)\mathbb{E}[\|s\|^2 \mid x]}} = \sqrt{G} \cdot \frac{\sqrt{\mathrm{RV}(x)}}{\sigma(x)}. \qquad\qquad \square$$

## I.5. Low-SNR updates induce parameter drift

When updates carry no directional signal (zero mean), the parameter drifts away from initialization at a rate linear in the number of steps. This illustrates why sustained low-SNR updates are harmful even if they do not systematically push in a wrong direction.

**Theorem I.4** (Illustrative random-walk drift under zero-mean noise). *Consider SGD-style updates*
$$\theta_{t+1} = \theta_t + \eta\,\xi_t,$$
*where $\{\xi_t\}_{t\geq 0}$ are independent, $\mathbb{E}[\xi_t] = 0$, and $\mathbb{E}[\|\xi_t\|^2] = v < \infty$ for all $t$. Then for any $T \geq 1$,*
$$\mathbb{E}\big[\|\theta_T - \theta_0\|^2\big] = \eta^2\,T\,v.$$

*Proof.* Unrolling the recursion yields
$$\theta_T - \theta_0 = \eta \sum_{t=0}^{T-1} \xi_t.$$

Therefore,
$$\|\theta_T - \theta_0\|^2 = \eta^2 \left\|\sum_{t=0}^{T-1} \xi_t\right\|^2 = \eta^2 \left(\sum_{t=0}^{T-1} \|\xi_t\|^2 + 2\sum_{0\leq i<j\leq T-1} \langle\xi_i, \xi_j\rangle\right).$$

Taking expectation and using independence with $\mathbb{E}[\xi_t] = 0$,
$$\mathbb{E}\langle\xi_i, \xi_j\rangle = \langle\mathbb{E}[\xi_i], \mathbb{E}[\xi_j]\rangle = 0, \qquad i \neq j.$$

Hence the cross terms vanish and
$$\mathbb{E}\big[\|\theta_T - \theta_0\|^2\big] = \eta^2 \sum_{t=0}^{T-1} \mathbb{E}[\|\xi_t\|^2] = \eta^2\,T\,v,$$

where we used $\mathbb{E}[\|\xi_t\|^2] = v$ for all $t$. $\qquad\square$

## J. Template Mixing Reduces Input Dependence

If the policy's conditional distribution is contaminated by a prompt-independent component $q(z)$ with mixing weight $\alpha$, the resulting mutual information $I_\alpha(X; Z)$ contracts by at least a factor of $(1 - \alpha)$. This formalizes the intuition that even partial drift toward a shared template erodes input dependence.

**Lemma J.1** (Template mixing contracts mutual information). *Let $X \sim P(X)$ and $Z \mid X = x \sim p(z \mid x)$ with marginal $p(z) = \mathbb{E}_{x \sim P}[p(z \mid x)]$. Fix any prompt-independent distribution $q(z)$. For $\alpha \in [0, 1]$, define the mixed conditional and marginal*

$$p_\alpha(z \mid x) := (1 - \alpha)p(z \mid x) + \alpha q(z), \qquad p_\alpha(z) := (1 - \alpha)p(z) + \alpha q(z).$$

*Let $I_\alpha(X; Z)$ denote the mutual information under $p_\alpha(x, z) = P(x)p_\alpha(z \mid x)$. Then*

$$I_\alpha(X; Z) \leq (1 - \alpha) I(X; Z).$$

*Proof.* For any fixed $x$,

$$\mathrm{KL}\big(p_\alpha(\cdot \mid x) \,\|\, p_\alpha(\cdot)\big) = \mathbb{E}_{z \sim p_\alpha(\cdot|x)} \left[ \log \frac{p_\alpha(z \mid x)}{p_\alpha(z)} \right]$$
$$= \mathbb{E}_{z \sim p_\alpha(\cdot|x)} \big[ \log p_\alpha(z \mid x) - \log p_\alpha(z) \big].$$

Taking expectation over $x \sim P(x)$ gives

$$I_\alpha(X; Z) = \mathbb{E}_x \Big[ \mathrm{KL}\big(p_\alpha(\cdot \mid x) \,\|\, p_\alpha(\cdot)\big) \Big].$$

The same identity holds for $I(X; Z)$ with $p_\alpha$ replaced by $p$.

By joint convexity of $\mathrm{KL}(\cdot \| \cdot)$ (Cover & Thomas, 2006, Theorem 2.7.2), for any distributions $a, b, c, d$ and any $\alpha \in [0, 1]$,

$$\mathrm{KL}\big((1 - \alpha)a + \alpha b \,\|\, (1 - \alpha)c + \alpha d\big) \leq (1 - \alpha)\mathrm{KL}(a\|c) + \alpha\,\mathrm{KL}(b\|d).$$

Let $a = p(\cdot \mid x)$, $b = q$, $c = p(\cdot)$, and $d = q$. Since

$$p_\alpha(\cdot \mid x) = (1 - \alpha)p(\cdot \mid x) + \alpha q, \qquad p_\alpha(\cdot) = (1 - \alpha)p(\cdot) + \alpha q,$$

we obtain

$$\mathrm{KL}\big(p_\alpha(\cdot \mid x) \,\|\, p_\alpha(\cdot)\big) \leq (1 - \alpha)\mathrm{KL}\big(p(\cdot \mid x) \,\|\, p(\cdot)\big) + \alpha\,\mathrm{KL}(q\|q)$$
$$= (1 - \alpha)\mathrm{KL}\big(p(\cdot \mid x) \,\|\, p(\cdot)\big).$$

Averaging over $x \sim P(x)$ yields

$$\mathbb{E}_x \Big[ \mathrm{KL}\big(p_\alpha(\cdot \mid x) \,\|\, p_\alpha(\cdot)\big) \Big] \leq (1 - \alpha)\,\mathbb{E}_x \Big[ \mathrm{KL}\big(p(\cdot \mid x) \,\|\, p(\cdot)\big) \Big].$$

Using the identity $I(X; Z) = \mathbb{E}_x \big[ \mathrm{KL}(p(\cdot \mid x) \,\|\, p(\cdot)) \big]$ (and the analogous one for $I_\alpha$), we obtain

$$I_\alpha(X; Z) \leq (1 - \alpha) I(X; Z),$$

which proves the lemma. $\qquad\square$

*Remark J.2.* The continuity bound $f(\varepsilon)$ depends on $\log(|\mathcal{X}||\mathcal{Z}|)$, which can be extremely large for LLM token spaces. Therefore, this result should be understood as a qualitative guarantee that KL-closeness implies MI-closeness in principle, rather than a tight quantitative bound in practice.

## K. Filtering Reduces Gradient-Estimation MSE

### K.1. Setup

Consider $P$ groups indexed by $i \in \{1, \dots, P\}$. Group $i$ contains $G$ rollouts, and $\widehat{g}_i \in \mathbb{R}^d$ denotes the *group-level* gradient estimator (already averaged over the $G$ rollouts in the group). We model

$$\widehat{g}_i = g_i + \varepsilon_i, \qquad \mathbb{E}[\varepsilon_i] = 0, \qquad \mathbb{E}\|\varepsilon_i\|^2 = \sigma_i^2,$$

where $\{\varepsilon_i\}_{i=1}^P$ are independent across groups. For a kept set $S$ of groups, we write $n := |S|$ for the number of kept groups.

## K.2. Unfiltered vs. filtered estimators

Define the unfiltered batch estimator and its mean:

$$\widehat{\bar{g}} := \frac{1}{P} \sum_{i=1}^{P} \widehat{g}_i, \qquad \bar{g} := \frac{1}{P} \sum_{i=1}^{P} g_i.$$

Let $S \subseteq \{1, \ldots, P\}$ be the set of kept groups with $|S| = n$. Define the filtered estimator and its mean:

$$\widehat{\bar{g}}_S := \frac{1}{n} \sum_{i \in S} \widehat{g}_i, \qquad \bar{g}_S := \frac{1}{n} \sum_{i \in S} g_i.$$

By retaining only a subset of prompt groups, the filtered estimator's mean-squared error depends solely on the noise variances of the kept groups. Dropping high-noise (low-RV) groups directly lowers the estimation error.

**Theorem K.1** (MSE of the filtered estimator). $\widehat{\bar{g}}_S$ *is unbiased for* $\bar{g}_S$ *and satisfies*

$$\mathbb{E}\|\widehat{\bar{g}}_S - \bar{g}_S\|^2 = \frac{1}{n^2} \sum_{i \in S} \sigma_i^2.$$

*Proof.* By the setup, $\widehat{g}_i = g_i + \varepsilon_i$ with $\mathbb{E}[\varepsilon_i] = 0$, hence

$$\mathbb{E}[\widehat{g}_i] = g_i.$$

Therefore,

$$\mathbb{E}[\widehat{\bar{g}}_S] = \frac{1}{n} \sum_{i \in S} \mathbb{E}[\widehat{g}_i] = \frac{1}{n} \sum_{i \in S} g_i = \bar{g}_S.$$

Moreover,

$$\widehat{\bar{g}}_S - \bar{g}_S = \frac{1}{n} \sum_{i \in S} (\widehat{g}_i - g_i) = \frac{1}{n} \sum_{i \in S} \varepsilon_i.$$

Therefore,

$$\mathbb{E}\|\widehat{\bar{g}}_S - \bar{g}_S\|^2 = \frac{1}{n^2} \mathbb{E}\left\|\sum_{i \in S} \varepsilon_i\right\|^2$$

$$= \frac{1}{n^2} \left( \sum_{i \in S} \mathbb{E}\|\varepsilon_i\|^2 + \sum_{\substack{i,j \in S \\ i \neq j}} \mathbb{E}\langle \varepsilon_i, \varepsilon_j \rangle \right).$$

By independence and $\mathbb{E}[\varepsilon_i] = 0$, for $i \neq j$ we have

$$\mathbb{E}\langle \varepsilon_i, \varepsilon_j \rangle = \langle \mathbb{E}[\varepsilon_i], \mathbb{E}[\varepsilon_j] \rangle = 0,$$

so the cross terms vanish. Hence

$$\mathbb{E}\|\widehat{\bar{g}}_S - \bar{g}_S\|^2 = \frac{1}{n^2} \sum_{i \in S} \mathbb{E}\|\varepsilon_i\|^2 = \frac{1}{n^2} \sum_{i \in S} \sigma_i^2.$$

$\square$

**Remark (bias relative to the original objective).** While $\widehat{\bar{g}}_S$ is unbiased for the *filtered* mean gradient $\bar{g}_S$, it is generally biased for the *unfiltered* mean gradient $\bar{g}$ unless $S$ is chosen independently of $\{g_i\}$ or $g_i$ is constant across groups.

# L. Reward-Agnostic Regularizers and Update Dominance

## L.1. Setup

Similarly, fix a prompt $x$ and consider trajectories $z \sim \pi_\theta(\cdot \mid x)$ with reward $R(z; x)$ and baseline $b(x)$. Define the reward-driven (task) gradient

$$g_{\text{task}}(x) := \mathbb{E}[(R(z; x) - b(x)) \, s(z; x) \mid X = x], \qquad s(z; x) := \nabla_\theta \log \pi_\theta(z \mid x).$$

Let $g_{\mathrm{reg}}(x)$ denote an update component that is computed without multiplying the reward (or advantage), e.g.,

$$g_{\mathrm{reg}}(x) := \lambda_{\mathrm{KL}}\, g_{\mathrm{KL}}(x) + \lambda_{\mathrm{ent}}\, g_{\mathrm{ent}}(x),$$

where $g_{\mathrm{KL}}(x)$ and $g_{\mathrm{ent}}(x)$ are gradients of prompt-level distributional regularizers. We write the total expected update as

$$g_{\mathrm{total}}(x) = g_{\mathrm{task}}(x) + g_{\mathrm{reg}}(x).$$

To summarize relative influence, define the dominance ratio

$$\rho(x) := \frac{\|g_{\mathrm{reg}}(x)\|}{\|g_{\mathrm{task}}(x)\| + \|g_{\mathrm{reg}}(x)\|} \in [0, 1].$$

We refer to $g_{\mathrm{reg}}(x)$ as *reward-agnostic* since it does not use within-prompt reward differences to weight trajectories.

### L.2. Low-RV prompts amplify regularizer influence

When reward variance is small, the task gradient weakens (by Theorem I.2) while regularizer gradients remain largely flat across prompts. Consequently, the regularizer's share of the total update grows on low-RV prompts, formalizing why these prompts are more prone to input-agnostic drift.

By Theorem I.2, for any prompt $x$,

$$\|g_{\mathrm{task}}(x)\| \le \sqrt{\mathrm{RV}(x)}\,\sqrt{\mathbb{E}[\|s\|^2 \mid X = x]}.$$

Therefore the dominance ratio

$$\rho(x) = \frac{\|g_{\mathrm{reg}}(x)\|}{\|g_{\mathrm{task}}(x)\| + \|g_{\mathrm{reg}}(x)\|}$$

admits the lower bound

$$\rho(x) \ge \frac{\|g_{\mathrm{reg}}(x)\|}{\|g_{\mathrm{reg}}(x)\| + \sqrt{\mathrm{RV}(x)}\sqrt{\mathbb{E}[\|s\|^2 \mid X = x]}}.$$

In particular, if $\|g_{\mathrm{reg}}(x)\|$ and $\mathbb{E}[\|s\|^2 \mid X = x]$ vary slowly across prompts compared to $\mathrm{RV}(x)$, then smaller $\mathrm{RV}(x)$ implies larger $\rho(x)$, i.e., the total update is more strongly shaped by reward-agnostic regularizers on low-RV prompts.

## M. KL-Closeness to the Base Implies MI-Closeness

If the current policy stays uniformly close to a reference policy in KL divergence, then the mutual information $I(X; Z)$ between inputs and reasoning also remains close. This means strong KL constraints preserve—but do not necessarily increase—input dependence.

**Theorem M.1.** *To avoid measure-theoretic issues, assume $X$ is supported on a finite set $\mathcal{X}$ and $Z$ takes values in a finite set $\mathcal{Z}$. Let $P(X)$ be the prompt distribution and define*

$$P_\theta(x, z) := P(x)\pi_\theta(z \mid x), \qquad P_0(x, z) := P(x)\pi_0(z \mid x).$$

*If*

$$\sup_{x \in \mathcal{X}} \mathrm{KL}(\pi_\theta(\cdot \mid x) \,\|\, \pi_0(\cdot \mid x)) \le \varepsilon,$$

*then there exists $f(\varepsilon) \to 0$ as $\varepsilon \to 0$ such that*

$$\big|I_\theta(X; Z) - I_0(X; Z)\big| \le f(\varepsilon).$$

*Proof.* By the chain rule for KL divergence,

$$\mathrm{KL}(P_\theta(X, Z) \,\|\, P_0(X, Z)) = \mathbb{E}_{x \sim P}[\mathrm{KL}(\pi_\theta(\cdot \mid x) \,\|\, \pi_0(\cdot \mid x))].$$

Under the assumption $\sup_{x \in \mathcal{X}} \mathrm{KL}(\pi_\theta(\cdot \mid x) \,\|\, \pi_0(\cdot \mid x)) \le \varepsilon$, we obtain

$$\mathrm{KL}(P_\theta(X, Z) \,\|\, P_0(X, Z)) \le \varepsilon.$$

By Pinsker's inequality,

$$\|P_\theta(X, Z) - P_0(X, Z)\|_{\mathrm{TV}} \le \sqrt{\tfrac{1}{2}\mathrm{KL}(P_\theta(X, Z) \,\|\, P_0(X, Z))} \le \sqrt{\tfrac{\varepsilon}{2}} =: \delta.$$

Since $\|P_\theta(X, Z) - P_0(X, Z)\|_{\mathrm{TV}} \le \delta$ and $(X, Z)$ takes values in a finite alphabet $\mathcal{X} \times \mathcal{Z}$, the Fannes-Audenaert inequality

implies
$$\big|H_\theta(X, Z) - H_0(X, Z)\big| \leq \delta \log(|\mathcal{X}||\mathcal{Z}| - 1) + h_2(\delta),$$
where $H_\theta(\cdot)$ denotes entropy under $P_\theta$, and $h_2(\cdot)$ is the binary entropy. Moreover, total variation does not increase under marginalization, so
$$\|P_\theta(Z) - P_0(Z)\|_{\mathrm{TV}} \leq \delta,$$
and applying Fannes-Audenaert on the alphabet $\mathcal{Z}$ yields
$$\big|H_\theta(Z) - H_0(Z)\big| \leq \delta \log(|\mathcal{Z}| - 1) + h_2(\delta) \leq \delta \log(|\mathcal{X}||\mathcal{Z}| - 1) + h_2(\delta).$$

Finally, using $I(X; Z) = H(X) + H(Z) - H(X, Z)$ and noting that $P_\theta(X) = P_0(X) = P(X)$ (hence $H_\theta(X) = H_0(X)$),
$$\begin{aligned}
\big|I_\theta(X; Z) - I_0(X; Z)\big| &= \big|(H_\theta(Z) - H_0(Z)) - (H_\theta(X, Z) - H_0(X, Z))\big| \\
&\leq \big|H_\theta(Z) - H_0(Z)\big| + \big|H_\theta(X, Z) - H_0(X, Z)\big| \\
&\leq 2\Big(\delta \log(|\mathcal{X}||\mathcal{Z}| - 1) + h_2(\delta)\Big).
\end{aligned}$$

Thus we may take
$$f(\varepsilon) := 2\Big(\delta \log(|\mathcal{X}||\mathcal{Z}| - 1) + h_2(\delta)\Big), \qquad \delta := \sqrt{\tfrac{\varepsilon}{2}},$$
which satisfies $f(\varepsilon) \to 0$ as $\varepsilon \to 0$. $\qquad\square$

## N. Decomposing Changes in Input Dependence

**Definition N.1** (Entropy changes). Let $X$ be prompts and let $Z \sim \pi_\theta(\cdot \mid X)$ under the current policy, with reference policy $\pi_0$. Define the conditional-entropy and marginal-entropy changes
$$\Delta_{\mathrm{in}} := H_\theta(Z \mid X) - H_0(Z \mid X), \qquad \Delta_{\mathrm{marg}} := H_\theta(Z) - H_0(Z).$$

The change in mutual information decomposes as $\Delta I = \Delta_{\mathrm{marg}} - \Delta_{\mathrm{in}}$. If an intervention (e.g., an entropy bonus) increases within-prompt variability $H(Z \mid X)$ more than it increases the marginal diversity $H(Z)$, input dependence necessarily decreases.

**Theorem N.2.** *With $\Delta_{\mathrm{in}}$ and $\Delta_{\mathrm{marg}}$ defined above,*
$$I_\theta(X; Z) - I_0(X; Z) = \Delta_{\mathrm{marg}} - \Delta_{\mathrm{in}}.$$
*In particular, if $\Delta_{\mathrm{in}} \geq \Delta_{\mathrm{marg}} + \gamma$ for some $\gamma > 0$, then*
$$I_\theta(X; Z) \leq I_0(X; Z) - \gamma,$$
*and especially $I_\theta(X; Z) < I_0(X; Z)$ whenever $\Delta_{\mathrm{in}} > \Delta_{\mathrm{marg}}$.*

*Proof.* Using $I(X; Z) = H(Z) - H(Z \mid X)$,
$$\begin{aligned}
I_\theta(X; Z) - I_0(X; Z) &= \big(H_\theta(Z) - H_0(Z)\big) - \big(H_\theta(Z \mid X) - H_0(Z \mid X)\big) \\
&= \Delta_{\mathrm{marg}} - \Delta_{\mathrm{in}}.
\end{aligned}$$
The sufficient-condition statements follow by rearranging the inequality. $\qquad\square$

An entropy bonus acts directly on the per-prompt dispersion and increases $H_\theta(Z \mid X)$, but it does not explicitly encourage cross-prompt separation that would increase the marginal entropy $H_\theta(Z)$ by a comparable amount. Hence it is plausible that $\Delta_{\mathrm{in}}$ exceeds $\Delta_{\mathrm{marg}}$, in which case Theorem N.2 implies $I_\theta(X; Z)$ decreases.

Appendix L explains that when $\mathrm{RV}(x)$ is small, the task update can be weak, so reward-agnostic regularizers can have larger relative influence on the total update.

## O. GRPO Normalization Amplifies Noise at Low RV

GRPO-style normalization divides the advantage by $\sqrt{\mathrm{RV}(x)}$, which induces a $\mathrm{RV}(x)^{-1}$ noise amplification in the mean-squared error of the per-prompt gradient estimator.

*Table 6.* MI proxy family. All variants are derived from in-batch cross-scoring of reasoning traces against prompts, using matched (per-token log-prob under the true prompt) and marginal (per-token log-prob under the uniform prompt mixture) as base quantities. First-turn variants use only the first agent turn; trajectory variants sample across all turns.

| Type | Proxy | Formula | Notes |
|---|---|---|---|
| Discrete | Retrieval-Acc | $\frac{1}{PG}\sum_{i,k}\mathbf{1}[\arg\max_j \mathbf{L}_{i,k,j}=i]$ | Chance level $1/P$ under template collapse |
| | Recall@$k$ | $\frac{1}{PG}\sum_{i,k}\mathbf{1}[i\in\text{top-}k_j(\mathbf{L}_{i,k,j})]$ | $k\in\{2,4,8\}$ |
| Continuous (raw) | MI-Est | $\frac{1}{PG}\sum_{i,k}(\text{matched}_{i,k}-\text{marginal}_{i,k})$ | Per-token; approaches 0 under collapse |
| | MI-Seq-Est | $\frac{1}{PG}\sum_{i,k}\left(\mathbf{L}_{i,k,i}-\log\frac{1}{P}\sum_j e^{\mathbf{L}_{i,k,j}}\right)$ | Per-sequence; no length normalization |
| Continuous (z-score) | MI-ZScore | $\frac{1}{PG}\sum_{i,k}\frac{\text{matched}_{i,k}-\text{marginal}_{i,k}}{\sigma_{\text{batch}}+\epsilon}$ | Normalized by current-batch marginal std |
| | MI-ZScore-EMA | $\frac{1}{PG}\sum_{i,k}\frac{\text{matched}_{i,k}-\text{marginal}_{i,k}}{\sigma_{\text{EMA}}+\epsilon}$ | $\sigma_{\text{EMA}}^{(t)}=\alpha\,\sigma_{\text{EMA}}^{(t-1)}+(1-\alpha)\,\sigma_{\text{batch}}^{(t)}$ |

For a fixed prompt $x$, define the normalized advantage

$$\widetilde{A}(z;x):=\frac{A(z;x)}{\sqrt{\text{RV}(x)}}, \qquad A(z;x):=R(z;x)-b(x), \qquad b(x):=\mathbb{E}_{z\sim\pi_\theta(\cdot|x)}[R(z;x)].$$

Given $K$ i.i.d. rollouts $z_1,\ldots,z_K\sim\pi_\theta(\cdot\mid x)$, define

$$\widehat{g}_{\text{GRPO}}(x):=\frac{1}{K}\sum_{k=1}^K \widetilde{A}_k\, s_k, \qquad g_{\text{GRPO}}(x):=\mathbb{E}[\widetilde{A}\,s\mid X=x],$$

where $s_k=\nabla_\theta\log\pi_\theta(z_k\mid x)$.

Dividing the advantage by $\sqrt{\text{RV}(x)}$ causes the gradient estimator's variance floor to scale as $\text{RV}(x)^{-1}$, so prompts with small reward variance suffer disproportionately noisy updates under GRPO-style normalization.

**Proposition O.1** (GRPO variance floor). *Under Assumption I.1, the GRPO estimator satisfies*

$$\mathbb{E}\left[\left\|\widehat{g}_{\text{GRPO}}(x)-g_{\text{GRPO}}(x)\right\|^2\mid X=x\right]\geq\frac{1}{K}\cdot\frac{\sigma^2(x)}{\text{RV}(x)}\,\mathbb{E}[\|s\|^2\mid X=x].$$

*If $\sigma(x)=0$, the lower bound is zero and thus vacuous.*

This bound makes explicit that smaller $\text{RV}(x)$ yields a larger variance floor for the normalized estimator if all other factors are the same.

# P. Core Author Contributions

Zihan Wang contributed across the full project lifecycle, including conceptualization, codebase and environment development, formal analysis, experiments, figures and plots, paper writing, and project correspondence. Chi Gui and Xing Jin contributed to the key ideas, software infrastructure, experiments, plots, and paper writing. Licheng Liu primarily contributed to the formal analysis and theory, experiments, and participated in paper writing. Qineng Wang contributed to the key ideas, software infrastructure, figures and plots, experiments, and paper writing.

# Q. MI Proxy Family (extended table)

Table 6 below lists the full MI proxy family used in the main paper, including discrete and continuous variants.

# R. Filtering Strategy Ablations

### R.1. $P\times G$ rollout-budget sweep

Table 7 sweeps prompt batch size $P$ and group size $G$ on Sokoban (Qwen2.5-3B) at fixed rollout budget $K{=}128$. Configurations with $G\geq 4$ and SNR-Aware Filtering match or beat the $128{\times}1$ baseline at 26–41% lower per-step time.

**Compute overhead of group sampling.** SNR-Aware Filtering requires at least $G{=}2$ trajectories per prompt (group size) to estimate per-prompt RV. Since the total rollout budget is fixed at $K{=}128$ trajectories, varying the prompt batch size $P$ and group size $G$ is a repartitioning of that budget — all configurations incur identical rollout cost. Table 7 shows performance

*Table 7.* Sweep over prompt batch size $P$ and group size $G$ (trajectories per prompt) on Sokoban (Qwen2.5-3B). NF = no filtering; F = SNR-Aware Filtering ($\rho$=0.9). Total rollout budget is fixed at 128 trajectories across all configurations.

| $P$ **(prompts)** $\times$ $G$ **(traj/prompt)** | **Task Perf. (%)** | | | **Step Time (s)** | | | **VRAM (GB)** | | |
|---|---|---|---|---|---|---|---|---|---|
| | NF | F | $\Delta$ | NF | F | $\Delta\%$ | NF | F | $\Delta$ |
| $128 \times 1$ | 23.6 | – | – | 89.8 | – | – | 201.80 | – | – |
| $64 \times 2$ | 18.8 | 27.3 | +8.6 | 91.8 | 64.9 | $-29\%$ | 201.39 | 201.83 | +0.44 |
| $32 \times 4$ | 24.2 | 27.4 | +3.2 | 89.8 | 52.6 | $-41\%$ | 202.11 | 201.67 | $-0.44$ |
| $8 \times 16$ | 15.6 | 23.6 | +8.0 | 89.2 | 65.9 | $-26\%$ | 201.54 | 201.90 | +0.36 |

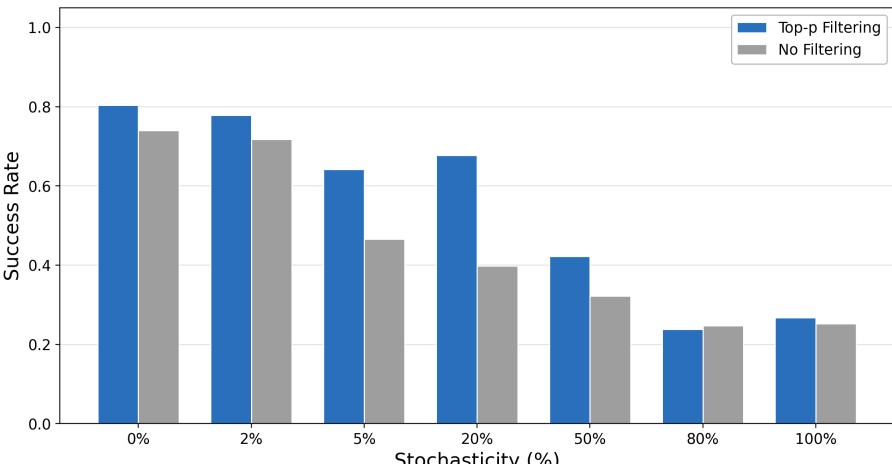

*Figure 11.* In FrozenLake, median success rates for both Top-p filtering (orange) and no filtering (gray) decrease as environment stochasticity increases from 0% to 100%. SNR-Aware Filtering maintains a clear advantage from 0% to 50% stochasticity, but the gap closes at 80%–100%, where high transition noise weakens reward variance as an informative signal proxy.

and wall-clock step time across configurations on Sokoban (Qwen2.5-3B). RV computation itself adds $<0.1\%$ of iteration time. With filtering ($\rho$=0.9), fewer groups enter gradient computation, reducing per-step time by 26–41%. Configurations with group size $G \geq 4$ and SNR-Aware Filtering match or outperform the $128 \times 1$ baseline, confirming that the gains come at no additional compute cost.

## S. Stress-Testing the SNR Mechanism

The SNR framing makes a concrete causal claim: template collapse is a gradient-level consequence of low reward variance, not a side effect of aggressive regularization or model capacity. We stress-test this claim with four questions: (1) Does directly controlling RV level causally drive performance and MI? (2) Does injecting environmental noise predictably weaken MI? (3) Do gains come from signal quality rather than prompt-distribution bias? (4) When does the filtering condition hold in practice? A positive answer to all four makes the SNR account difficult to dismiss.

**Quartile ablation provides direct causal evidence.** To move beyond correlation between RV and performance, we run a controlled intervention. We sort all prompt groups by within-prompt RV, divide them into four quartiles (Q1 = highest, Q4 = lowest), and train four separate runs — each updating on one quartile only, all other settings fixed (Table 8). Task performance and MI degrade monotonically from Q1 to Q4. Combined with Theorem G.1 ($\|g_{\text{task}}\| \leq \sqrt{\text{RV}}$), this establishes the full causal chain: reward variance $\rightarrow$ gradient quality $\rightarrow$ input-dependent reasoning.

**Controlled noise injection weakens MI.** We run a direct intervention: varying environmental stochasticity and asking whether MI declines *predictably* in response. As environment and policy randomness increases, task return drops, conditional entropy rises, and $\widehat{I}(X; Z)$ decreases monotonically (Figure 11). This is the expected consequence of the SNR chain. Additional noise inflates within-prompt return variance in a signal-free way, diluting the advantage estimates that task gradients depend on. Importantly, the filter's advantage also attenuates at very high noise (80–100%), which is itself informative: when the environment is so stochastic that even high-effort prompts yield noisy rewards, RV loses its discriminative power. The mechanism predicts exactly this boundary condition.

*Table 8.* Quartile ablation on Sokoban (Qwen2.5-3B, $P{=}8$, $G{=}16$, keeping 25% of prompts per step). Task performance and MI degrade monotonically from Q1 to Q4.

| Quartile | RV Range | Task Perf (%) | MI Proxy | Entropy |
|---|---|---|---|---|
| Q1 (highest RV) | [4.4–5.6] | 21.1 | 0.95 | 2.02 |
| Q2 | [1.5–4.2] | 19.5 | 0.93 | 1.53 |
| Q3 | [0.0–0.2] | 10.7 | 0.81 | 1.41 |
| Q4 (lowest RV) | [0.0–0.1] | 11.0 | 0.73 | 1.87 |

**Prompt-level filtering outperforms trajectory-level filtering.** The gains from SNR-Aware filtering could come from selecting discriminative prompts, or from discarding hard/noisy trajectories. We disentangle these with a trajectory-level baseline: we keep all prompts but retain only the top-8 and bottom-8 trajectories per prompt by reward, preserving the prompt distribution while improving per-prompt SNR (Table 9). Trajectory-level filtering improves over no filtering. However, prompt-level SNR-Aware Filtering outperforms it by a wider margin. Within a naturally low-RV prompt, forcing within-prompt variance by sub-selecting trajectories amplifies noise. Selecting prompts that naturally produce discriminative signals is more effective.

*Table 9.* Trajectory-level vs. prompt-level filtering on Sokoban (Qwen2.5-3B). Prompt-level SNR-Aware Filtering provides the largest gains; trajectory-level filtering confirms that the benefit is not due to prompt-distribution bias.

| Method | Prompts Used | Traj/Update | Task Perf (%) | MI Proxy |
|---|---|---|---|---|
| No filter | 8/8 | 128 | 12.9 | 0.83 |
| Prompt-level RV ($\rho{=}0.9$) | 3.2/8 | 50.6 | 23.6 | 1.80 |
| Trajectory-level | 8/8 | 64 | 16.8 | 0.20 |

**When does SNR-Aware Filtering help?** Finally, we find SNR-Aware Filtering improves performance better when cross-prompt RV heterogeneity is large enough to separate signal-rich from noise-only prompts. We find the metric Std(RV)/Mean(RV), computable from a single rollout batch, can effectively predict this (Table 10). When the ratio is high, the per-prompt RV distribution is bimodal and filtering cleanly separates signal from noise. When the ratio is near zero, all prompts carry similar RV and filtering discards data uniformly, like FrozenLake GRPO ($\Delta{=}{-}5.0\%$, ratio 0.33). This ratio is a cheap diagnostic which can be done before training.

*Table 10.* Per-setting RV statistics and filtering effectiveness. Std/Mean of RV predicts whether SNR-Aware Filtering helps: high ratio means bimodal RV and effective filtering; low ratio means uniform RV and random discarding.

| Setting | Filter $\Delta$ | P | G | RV Mean | RV Std | RV Var | RV Min | RV Max | Std/Mean |
|---|---|---|---|---|---|---|---|---|---|
| Sokoban, 14B | +4.6% | 8 | 8 | 2.24 | 2.88 | 8.32 | 0.10 | 6.00 | 1.29 |
| Sokoban, 3B | +3.2% | 32 | 4 | 2.49 | 2.89 | 8.35 | 0.05 | 6.52 | 1.16 |
| FrozenLake, 3B (GRPO) | −5.0% | 32 | 8 | 0.54 | 0.18 | 0.03 | 0.22 | 0.76 | 0.33 |

**How filtering adapts as training progresses?** With the four predictions confirmed, we can now characterize how SNR-Aware Filtering behaves over the full training trajectory. Figure 12 tracks the effective kept ratio $\rho_{\text{eff}}$ and zero-variance prompt count over training. Both move in the expected direction: as the policy improves and converges, more prompts yield near-identical rollout rewards (zero-variance count rises), and the filter responds by becoming more selective (kept ratio falls). This automatic tightening is precisely what a fixed strategy like Top-$k$ with constant $k$ cannot replicate. It would continue absorbing gradient budget from uninformative prompts even as signal quality deteriorates.

**Reward collapse is visible at the distribution level.** Figure 13 provides a complementary view of the same dynamics, tracking prompt-level reward distributions across early, mid, and late training in Sokoban. The shift is systematic: the hard portion shrinks as the policy improves, the mixed portion expands, and overall prompt-level variance collapses toward the late stages. This distribution-level signature mirrors the gradient-level story. Late training is not simply "easier" for the policy; it is a regime where reward variation has been compressed to the point that gradient updates carry progressively less task-discriminative information.

**Format validity cannot substitute for content-sensitive diagnostics.** One might hope that a coarser signal (whether the model's output follows the required format) could serve as a collapse indicator without the overhead of MI estimation. Figure 14 shows this does not hold: format validity is largely decoupled from collapse, with runs maintaining near-perfect

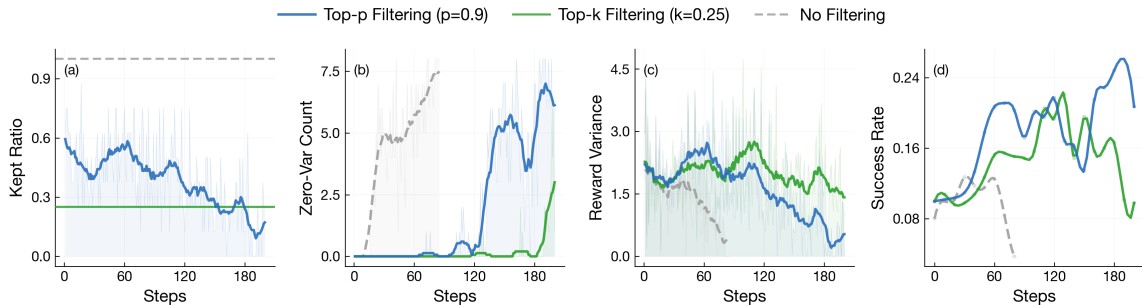

*Figure 12.* Effective kept ratio and zero-variance prompt count, showing adaptive selection pressure as variance collapses during training.

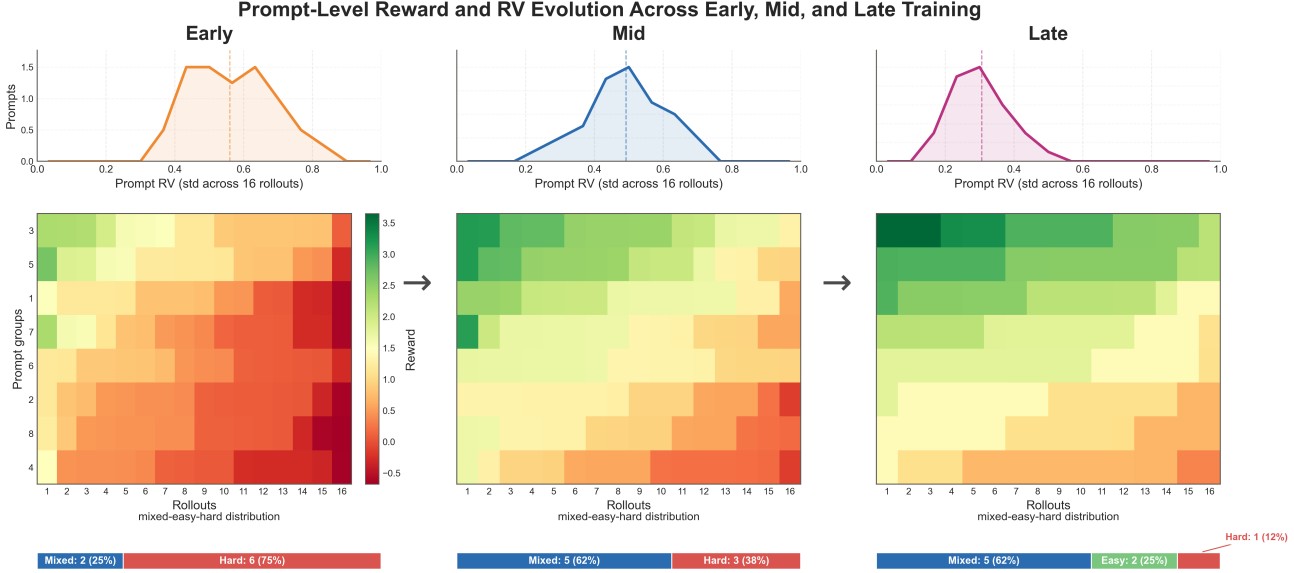

*Figure 13.* Prompt-level reward distribution across training phases, showing RV collapse as prompts shift toward uniform reward structures.

validity while exhibiting low MI. Structural correctness and semantic input-dependence are separate dimensions. This reinforces the need for content-sensitive diagnostics, and explains why the MI proxy provides signal that format-based checks miss.

RV is largely orthogonal to entropy and response length, which explains why entropy-based stabilizers cannot prevent template collapse. Reward variance correlates weakly with conditional entropy (Spearman $-0.14$) and response length $(0.12)$, while correlating strongly with task reward $(0.63)$. RV therefore targets a distinct axis of update quality rather than surface statistics, making it a complementary control knob to KL and entropy regularization. Figure 12 further shows that the effective kept ratio adapts over training: as more prompts drift toward near-zero RV, the filter automatically concentrates gradient updates on the shrinking pool of still-informative prompts.

**What is the relationship between SNR-Aware Filtering and KL/entropy tuning stabilization?** When training RL agents, practitioners typically tune KL penalty and entropy regularization coefficients to maintain training stability and prevent mode collapse. However, these interventions primarily control within-input diversity $(H(Z \mid X))$ and cannot directly address the signal-to-noise imbalance that drives template collapse. Even with carefully tuned regularization, if most prompts have low reward variance, the task gradient remains weak and regularization forces still dominate the update direction.

SNR-Aware Filtering is complementary: it selects high-signal prompts at each iteration, directly boosting the fraction of task-discriminative gradient in each update. This acts as a signal-enhancement mechanism rather than a noise-control mechanism. We provide a detailed empirical comparison of KL tuning, entropy tuning, and SNR-Aware Filtering in Section 5.1, showing that the three interventions move training dynamics along different axes (Figure 8).

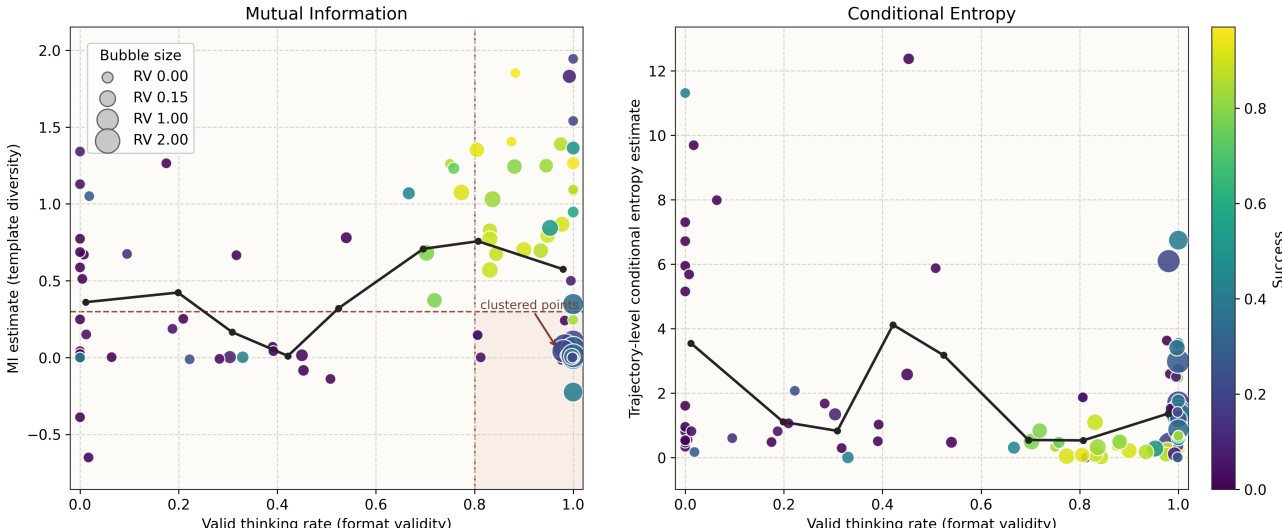

*Figure 14.* Format validity versus MI and entropy diagnostics, showing that high validity does not guarantee high input dependence.

# T. Extended Limitations

The SNR decomposition assumes task-signal and regularization noise separate cleanly, though they may couple through gradient accumulation in practice. All experiments are single-agent; how template collapse propagates in multi-agent RL remains open. A capable model could game the filtering criterion by artificially inflating reward variance, a risk worth monitoring over long training horizons. The method requires reward variance to be a reliable signal proxy, which degrades in sparse or noisy reward environments. Aggressive filtering may narrow exploration coverage; the kept mass requires per-task tuning.

# U. Core Author Contributions

Zihan Wang contributed across the full project lifecycle, including conceptualization, codebase and environment development, formal analysis, experiments, figures and plots, paper writing, and project correspondence. Chi Gui and Xing Jin contributed to the key ideas, software infrastructure, experiments, plots, and paper writing. Licheng Liu primarily contributed to the formal analysis and theory, experiments, and participated in paper writing. Qineng Wang contributed to the key ideas, software infrastructure, figures and plots, experiments, and paper writing.

