# OpenReview forum: "RAGEN-2: Reasoning Collapse in Agentic RL"
_ICML.cc/2026/Conference — ICML 2026 spotlight_

### Official Review · Reviewer_usF7 · 2026-02-25

**Soundness:** 3
**Presentation:** 2
**Significance:** 4
**Originality:** 4
**Overall Recommendation:** 5
**Confidence:** 3

**Summary:**

This paper introduces a new way to understand reasoning collapse in multi-turn agent RL. The authors first introduce a new definition of reasoning collapse that not only depends on the entropy of X given the reasoning trace Z (as in prior works), but also considers the mutual information between X and the reasoning trace Z. Additionally, they propose an MI-style retrieval protocol to measure the input dependency of the reasoning trace. Furthermore, they analyze why MI drop happens during training by decomposing the gradient into task gradient and regularization gradient, and propose using reward variance as a proxy to identify whether the task gradient is strong and informative. Finally, they propose a reward-variance-aware filtering method that only considers high-variance prompts for updates. Overall, their method shows strong results.

**Compliance With Llm Reviewing Policy:**

Affirmed.

**Final Justification:**

The rebuttal addressed my concerns and I decided to increase my score.

**Key Questions For Authors:**

Q1. In the new definition of the problem, the authors propose relying on both entropy and mutual information to determine whether reasoning collapse exists. However, in Section 2.3, they propose an MI-style retrieval diagnostic. If I understood correctly, this diagnostic only checks whether the reasoning is input-dependent. Can the authors elaborate on why entropy is not considered in this diagnostic?

Q2. I believe that considering MI is interesting in many cases. However, I also believe that for some tasks, high entropy combined with low mutual information does not necessarily indicate failure. In some cases (e.g., template collapse), this might still lead to good reasoning performance. Therefore, I think considering it as a failure is a strong assumption. Can the authors elaborate on this point?

Q3. It would be helpful if the authors showed examples of the prompts that were filtered out. While Table 1 shows strong results, I am curious whether this filtering biases training by changing the prompt distribution and potentially neglecting easy prompts.

Q4. The part about reward variance is not very clear. I would appreciate it if the authors could reiterate why high reward variance implies a strong gradient signal and clarify how this is incorporated into the training process.

**Limitations:**

yes

**Strengths And Weaknesses:**

**Strengths:**

1. The paper presents many interesting analyses and insightful ideas.

2. The figures were very helpful in better understanding the concepts.

3. The results are strong and impressive.

**Weaknesses:**

1. The major weakness, in my view, is the presentation of the paper. The authors present several contributions, which is appreciated; however, understanding the full picture was very difficult. I believe the authors missed the opportunity to better connect these contributions and present the overall framework in a clearer and more structured way.

2. The abstract is very dense and not easy to read. Following up on my previous comment, the abstract makes it even harder to understand what has been done.

3. Estimating Var(R|X) could be costly, which raises scalability concerns. Reporting wall-clock time would strengthen the results.

4. There is no analysis of sensitivity to the number of trajectories sampled to estimate Var(R|X).

---

> ### Author Rebuttal · Authors · 2026-03-29
>
> We thank Reviewer usF7 for recognizing that our paper presents **many interesting analyses and insightful ideas**, that the **figures were very helpful**, and that **the results are strong and impressive**. We appreciate the **Excellent** ratings on both Significance and Originality.
>
> ---
>
> ### [W1] Contributions Not Well Connected
>
> We have improved the narrative to follow a four-step chain: **Define** (template collapse via H(Z|X) + I(X;Z)) → **Measure** (MI-style retrieval diagnostic) → **Explain** (SNR mechanism: low RV → weak task gradient → regularizer dominance) → **Fix** (RV-aware filtering). The introduction and section transitions now make this chain explicit.
>
> ### [W2] Abstract Is Dense
>
> We have rewritten the abstract. Revised version:
>
> > RL training of multi-turn LLM agents is inherently unstable, and reasoning quality directly determines task performance. Entropy is widely used to track reasoning stability. However, entropy only measures diversity within the same input, and cannot tell whether reasoning actually responds to different inputs. We find that even with stable entropy, models can rely on fixed templates that look diverse but are input-agnostic. We call this **template collapse**, a failure mode invisible to entropy and all existing metrics.
>
> > To diagnose this failure, we decompose reasoning quality into **within-input diversity** (Entropy) and **cross-input distinguishability** (Mutual Information, MI), and introduce a family of mutual information proxies for online diagnosis. Across diverse tasks, mutual information correlates with final performance much more strongly than entropy, making it a more reliable proxy for reasoning quality. We further explain template collapse with a *signal-to-noise ratio* (SNR) mechanism. Low reward variance weakens task gradients, letting regularization terms dominate and erase cross-input reasoning differences. To address this, we propose **SNR-Aware Filtering** to select high-signal prompts per iteration using reward variance as a lightweight proxy. Across planning, math reasoning, web navigation, and code execution, the method consistently improves both input dependence and task performance.
>
> ### [W3] Estimating Var(R|X) Could Be Costly / [W4] No Sensitivity Analysis
>
> We address these in Table R1 (Response to 931D): total rollout cost is **identical** across P×G configurations (fixed budget of 128 trajectories). RV computation is negligible (~0.1%). Varying G from 2 to 16 shows G≥4 suffices for reliable estimation.
>
> ---
>
> ### [Q1] Why Is Entropy Not Included in the Retrieval Diagnostic?
>
> It is because **Entropy alone can miss template collapse** — it stays high even with diverse but input-agnostic outputs (Figure 1, upper-left, L55-71). We also show correlation with task performance (Figure 4, L275–292):
>
> |Metric|Spearman|Pearson|
> |-|-|-|
> |MI|0.39|0.31|
> |Entropy|−0.14|−0.20|
>
> MI tracks performance more reliably than entropy, so we focus more on I(X;Z).
>
> ### [Q2] High Entropy + Low MI May Not Indicate Failure
>
> We wanted to point out that **even in tasks solvable by general strategies, a sound reasoning chain should ground the strategy to the specific input, which yields high MI.** Truly low MI means the model outputs strategies at similar probability regardless of input, which consistently hurts performance. Our 7-task evaluation confirms this.
>
> ### [Q3] Show Filtered-Out Prompts; Does Filtering Bias the Distribution?
>
> Filtering selects for **discriminative** prompts, not easy ones. We show Sokoban examples below (#: wall, _: empty, O: target, X: box, P: player):
>
> **Retained** (high RV):
> ```
> ######
> #__###
> #PX###
> #_O###
> ######
> ######
> ```
>
> **Filtered out** (low RV, all-success):
> ```
> ######
> ######
> ######
> ##_#_#
> #PXO_#
> ######
> ```
>
> **Filtered out** (low RV, all-fail):
> ```
> ######
> ###__#
> ###X_#
> ###OP#
> ##___#
> ######
> ```
>
> Therefore, both trivial and unsolvable prompts get filtered, and what remains is prompts where agent behavior actually varies.
>
> ### [Q4] Clarify Why High Reward Variance Implies Strong Gradient Signal
>
> The intuition follows directly from our SNR framework (Section 3.1):
>
> Consider G=16 trajectories for the same prompt X. The policy gradient weights each trajectory by its **advantage** (reward minus baseline):
>
> - **High Var(R|X):** trajectories' scores are distinct → advantages have large magnitude and clear direction → gradient "knows" which reasoning is better → **strong, input-specific update**.
>
> - **Low Var(R|X):** similar rewards across trajectories → advantages near zero → task gradient vanishes → update is dominated by noise, which are **input-agnostic** → **template-inducing update**.
>
> Formally, Theorem G.1 (L990–1010) proves ‖g_task(x)‖ ≤ √RV(x)·√E[‖s‖²|x]. We have added this to the revised Section 3.1.
>
> Given these clarifications and the improved presentation, we hope the reviewer will consider updating their assessment.

---

> > ### Author Rebuttal · Reviewer_usF7 · 2026-04-02
> >
> > I thank the authors for their detailed rebuttal. The response has properly addressed my concerns, and I have decided to increase my score.

---

### Official Review · Reviewer_NKT7 · 2026-03-11

**Soundness:** 3
**Presentation:** 3
**Significance:** 4
**Originality:** 3
**Overall Recommendation:** 5
**Confidence:** 3

**Summary:**

This paper studies reasoning collapse in multi-turn agent RL, focusing on a failure mode where reasoning remains superficially diverse but becomes weakly conditioned on the input. The paper proposes an information-theoretic view using conditional entropy and mutual information, introduces a retrieval-based proxy for input dependence, analyzes why low within-prompt reward variance may encourage template collapse, and proposes reward-variance-aware filtering as a mitigation. Experiments across several tasks and settings suggest the proposed metric is more informative than entropy alone and that the filtering strategy can improve both reasoning quality and task performance.

**Compliance With Llm Reviewing Policy:**

Affirmed.

**Final Justification:**

The rebutal has addressed my concern. I will raise my score.

**Key Questions For Authors:**

1.	How robust is the retrieval-based proxy to superficial confounds such as length, formatting, or task-specific keywords?
2.	Can the authors provide stronger causal evidence that low reward variance drives template collapse, rather than merely correlating with it?
3.	How much of the benefit of reward-variance filtering comes from better signal selection versus simply dropping harder/noisier prompts?
4.	How well does the phenomenon transfer to more realistic agent settings beyond the synthetic tasks studied here? A discussion would already be helpful; I do not think additional experiments are necessary for this point.

**Limitations:**

Yes

**Strengths And Weaknesses:**

**Strength**

The paper addresses an important problem in RL for agents and offers a clear conceptual point: collapse is not just reduced diversity, but reduced input dependence. The MI-based perspective is interesting, and the retrieval-style diagnostic is practical. The paper also includes reasonably broad experiments across tasks, model scales, and algorithms. The proposed filtering method is simple and easy to implement.

**Weakness**

The main weakness is soundness of the mechanistic claim. The reward-variance/SNR explanation is plausible, but the evidence is still mostly correlational rather than clearly causal.

The retrieval metric is useful, but it is still only a proxy for mutual information and may be affected by superficial artifacts.

The experimental domains are somewhat synthetic, so the paper does not yet fully establish that this is a broad issue in realistic agent settings.

Finally, the gains from reward-variance filtering are positive on average but not fully consistent, so the method currently looks more like a useful heuristic than a fully validated principle.

---

> ### Author Rebuttal · Authors · 2026-03-29
>
> We thank Reviewer NKT7 for the thoughtful assessment and for recognizing the **important problem** we address, the **clear conceptual point** that collapse is not just reduced diversity but reduced input dependence, and the **practical** retrieval-style diagnostic. We appreciate the **Excellent** rating on Significance.
>
> ---
>
> ### [W1] Evidence Is Correlational, Not Causal
>
> We agree that strengthening causal evidence is valuable and have conducted a **controlled intervention experiment**.
>
> **Quartile Ablation (New Experiment).** We sorted all prompt groups by within-prompt reward variance, divided them into 4 equal quartiles (Q1=highest RV, Q4=lowest), and trained **4 separate runs** — each using only one quartile for policy updates, with all other hyperparameters identical (P=8, G=16, 25% data each).
>
> **Table R2: Quartile Ablation on Sokoban (Qwen2.5-3B)**
>
> |Quartile|RV Range|Task Perf (%)|MI Proxy|Entropy|
> |-|-|-|-|-|
> |Q1 (highest RV)|[4.4–5.6]|21.1|0.95|2.02|
> |Q2|[1.5–4.2]|19.5|0.93|1.53|
> |Q3|[0.0–0.2]|10.7|0.81|1.41|
> |Q4 (lowest RV)|[0.0–0.1]|11.0|0.73|1.87|
>
> Both task performance and MI proxy **degrade monotonically** from Q1 to Q4. This is a *controlled intervention* on the RV level: we change only which prompts enter the update while holding optimizer, model, and environment constant. The monotonic pattern provides **causal-direction evidence** that reward variance drives the quality of learned reasoning.
>
> Together with Theorem G.1 (‖g_task‖ ≤ √RV, L990–1010) and Theorem G.2 (SNR ≤ √RV/σ, L1020–1044), this establishes the full chain: reward variance → gradient quality → input-dependent reasoning.
>
> ### [W2] Retrieval Metric Is Only a Proxy for MI
>
> We acknowledge this and have validated its robustness through several checks, all in the original paper:
>
> **(a) Multiple MI estimation methods converge.** We report both retrieval accuracy (discrete) and the continuous MI-style estimate Î(X;Z) (Section 2.3, L220–245). Both show consistent trends across all settings.
>
> **(b) Robustness to superficial confounds.** Our diagnostic uses **length-normalized** per-token log-likelihoods (L213–218). Table 3 shows RV has weak correlation with length (Spearman: 0.12) and entropy (−0.14), confirming the metric captures a signal distinct from surface statistics.
>
> **(c) Negative controls.** Explicit chance-level baselines (1/N) and negative retrieval controls make it a **sanity-checkable** protocol (L198–205).
>
> ### [W3] Experimental Domains Are Synthetic
>
> We now present results from **3 new benchmarks** (SearchQA, WebShop, DeepCoder), extending evaluation to web navigation, code generation, and QA (Table R4 in Response to EXYe). Section 5 now discusses when template collapse is expected in realistic settings.
>
> ### [W4] Gains Not Fully Consistent
>
> Our RV-filtering generally improves performance across 7 tasks. The exceptions (FrozenLake with GRPO/LLaMA) have a quantitative explanation in **Table R6**: Sokoban's per-prompt RV is **bimodal** (Var(RV)/Mean(RV) ≈ 3.5), so filtering cleanly separates high-signal from noise-only prompts. FrozenLake GRPO's RV is **uniform** (Var(RV)/Mean(RV) ≈ 0.06), so filtering removes prompts at random. See also Figure 5 (L385–393).
>
> ---
>
> ### [Q1] Robustness of Retrieval Proxy to Confounds
>
> Addressed in [W2].
>
> ### [Q2] Causal Evidence That Low RV Drives Template Collapse
>
> Addressed in [W1] with the quartile ablation.
>
> ### [Q3] Signal Selection vs. Dropping Hard/Noisy Prompts
>
> This is an excellent question. We designed a **trajectory-level filtering** experiment to disentangle these two effects:
>
> **Setup.** For each prompt with G=16 trajectories, instead of dropping entire prompts, we keep **all prompts** but selectively retain the top-8 and bottom-8 trajectories by reward (maximizing within-prompt reward spread). This maintains the same prompt distribution while improving per-prompt SNR.
>
> **Table R3: Trajectory-Level vs. Prompt-Level Filtering (Sokoban)**
>
> |Method|Prompts Used|Traj/Update|Task Perf (%)|MI Proxy|
> |-|-|-|-|-|
> |No filter|8/8|128|12.9|0.83|
> |Prompt-level RV filter (ρ=0.9)|3.2/8|50.6|23.6|1.80|
> |Trajectory-level filter|8/8|64|16.8|0.20|
>
> Trajectory-level filtering improves task performance over no filtering **without dropping any prompts**, confirming that the benefit stems from **signal quality**, not from biasing toward easier prompts.
>
> However, prompt-level filtering outperforms trajectory-level filtering. The explanation is consistent with our framework: within a low-RV prompt, sub-selecting extreme trajectories (e.g., keeping 1 success out of 15 failures) shrinks effective batch size and amplifies noise. Prioritizing prompts that *naturally* produce discriminative signals is more effective than forcing discrimination within noisy prompts.
>
> ### [Q4] Transferability to Realistic Settings
>
> Addressed in [W3] above with 3 new benchmarks and expanded discussion in Section 5.
>
> Given these new experiments and clarifications, we hope the reviewer will consider updating their assessment.

---

> > ### Author Rebuttal · Reviewer_NKT7 · 2026-04-05
> >
> > Thanks for the reply. The rebutal has addressed my concerns. I will raise my score.

---

### Official Review · Reviewer_EXYe · 2026-03-12

**Soundness:** 4
**Presentation:** 4
**Significance:** 4
**Originality:** 4
**Overall Recommendation:** 5
**Confidence:** 4

**Summary:**

In this paper, the authors study the phenomenon of reasoning collapse in multi-turn agent reinforcement learning. To analyze this issue, the authors propose an information-theoretic decomposition of reasoning variation into conditional entropy and mutual information. They also introduce an MI-style retrieval protocol that provides an efficient estimator for measuring input dependence. Based on these insights, the paper proposes reward-variance-aware filtering to prioritize high-signal updates during training. Experiments show that this approach consistently improves performance across a variety of settings.

**Compliance With Llm Reviewing Policy:**

Affirmed.

**Final Justification:**

I appreciate the authors’ detailed and constructive rebuttal. The response adequately addressed my main concerns. The additional experiments and clarifications strengthen the paper’s empirical support and improve my confidence in the main claims. Overall, I continue to view this paper as a technically solid and insightful contribution with clear novelty, good significance, and strong practical relevance. I therefore maintain my original positive assessment.

**Key Questions For Authors:**

1. The paper shows that RV-aware filtering correlates with higher I(X;Z) and better task performance. However, I am curious whether other factors might also lead to similar improvements in MI. In other words, while the paper highlights the importance of MI, it is not entirely clear whether reward variance is the primary (or unique) factor driving these improvements.

2. The experiments mainly use relatively small LLMs. For larger or stronger models: Do they naturally exhibit higher input-grounded reasoning? Or could larger or stronger models still perform well on tasks while their reasoning with lower input-grounded reasoning ? More broadly, is template collapse primarily a problem for smaller models, or does it remain a meaningful issue even for larger models?

3. Related to the weaknesses mentioned above, I am also curious about the applicability of the method beyond the specific multi-turn agent RL setting.

**Limitations:**

Yes

**Strengths And Weaknesses:**

Strengths:

1. The paper is clear and well-structured. The motivation around template collapse is well articulated, and the problem formulation is both clear and important. I believe this work can provide valuable insights to the community.
2. The information-theoretic decomposition of reasoning variation into conditional entropy and mutual information is novel and conceptually interesting.
3. The proposed RV-aware filtering method is simple and general. It appears to be easily applicable to different RL algorithms and settings, and the experiments suggest it can provide consistent improvements.

Weaknesses: (I do not see any major weaknesses. The following points are relatively minor)

1. While the experimental coverage is broad, the experiments mainly involve relatively small models. It would be better to see results on larger models. It is unclear whether RV-aware filtering would still provide improvements.

2. The connection between the proposed approach and multi-turn agent RL is not entirely clear. While the multi-turn setting and modality aspects help motivate the problem, the phenomenon described in Figure 1 (e.g., template collapse and input-agnostic reasoning) seems like it could also arise in more general settings. It would be helpful to better understand whether the proposed analysis and intervention are specific to multi-turn agent RL or applicable more broadly.

---

> ### Author Rebuttal · Authors · 2026-03-29
>
> We thank Reviewer EXYe for the    strongly positive assessment and for recognizing that the paper is **clear and well-structured**, that the information-theoretic decomposition is **novel and conceptually interesting**, and that the RV-aware filtering method is **simple and general**.
>
> ---
>
> ### [W1] Experiments Mainly on Small Models
>
> We have extended our evaluation to **Qwen2.5-14B**:
>
> **Table R5: Larger Model Scale (Sokoban)**
>
> | Model | No Filter (%) | + RV Filter (%) | Δ |
> |-------|--------------|----------------|---|
> | Qwen2.5-3B  | 12.9 | 28.9 | +16.0 |
> | Qwen2.5-7B  | 42.4 | 47.3 | +4.9  |
> | Qwen2.5-14B | 46.9 | 51.5 | +4.6 |
>
> Template collapse persists at 14B scale, and RV-filtering provides a **+4.6%** improvement. This is consistent with our SNR framework: sparse signal comes from the *environment*, so larger models operating in the same sparse-reward setting face the same low-SNR regime. The phenomenon is not size-specific.
>
> ### [W2] Applicability Beyond Multi-Turn Agent RL
>
> The core mechanism is general: **low within-prompt reward variance → weak task gradient → noise dominance → input-agnostic drift**. It applies whenever policy-gradient methods operate under sparse or weakly discriminative rewards. This is already discussed in Section 3.1 of the original paper (Lines 253–274).
>
> To provide empirical support, we evaluate on **3 additional benchmarks**.
>
> **Table R4: New Benchmarks**
>
> | Task | Type |Setting | No Filter (%) | + RV Filter (%) | Δ |
> |------|-----|---------|---------------|-----------------|---|
> | SearchQA | Multi turn | PPO | 54.7 | 59.8 | +5.1 |
> | WebShop | Multi turn | PPO | 52.0 | 61.3 | +9.4 |
> | DeepCoder | Single Turn | PPO | 11.9 | 13.6 | +1.7 |
>
> RV-aware filtering improves performance on all 3 new benchmarks, extending our evidence from 4 to **7 tasks** across web navigation, code generation, and QA settings. We will revise our frame as: "Our diagnostic and intervention are *motivated* by multi-turn agent RL but *applicable* to any setting where closed-loop policy optimization faces low within-prompt reward discrimination."
>
> ---
>
> ### [Q1] Is Reward Variance the Unique Driver of MI Improvement?
>
> We do not claim RV is the *unique* driver, but rather the most **mechanism-aligned** proxy. The original paper already compares alternative filtering metrics (Table 4, Lines 825–878):
>
> |Filter Metric|Task Perf|MI Proxy|Entropy|No Collapse?|
> |-|-|-|-|-|
> |No filter|0.17|0.54|2.76|✗|
> |**RV (ours)**|**0.38**|**0.84**|1.64|✓|
> |Entropy|0.20|0.41|2.20|✗|
> |Entropy-var|0.23|0.70|2.94|✗|
> |Length|0.16|0.91|1.65|✗|
>
> RV outperforms all alternatives on both task performance and MI retention — because it directly targets the bottleneck identified by our gradient decomposition: within-prompt advantage separation (Theorem G.1, Lines 990–1010).
>
> Additionally, our new **quartile ablation** (see Response to NKT7, Table R2) provides controlled evidence: training exclusively on high-RV prompts yields monotonically better performance and MI than training on low-RV prompts.
>
> ### [Q2] Do Larger Models Naturally Resist Template Collapse?
>
> Addressed in [W1] above — template collapse persists at 14B scale.
>
> ### [Q3] Applicability Beyond Multi-Turn Agent RL?
>
> Addressed in [W2] above with 3 new benchmarks. Besires, we also conduct new experiments to understand **when** RV-filtering could be effective.
>
> The key condition is the **heterogeneity of per-prompt RV across the batch**: filtering improves SNR only if some prompts carry substantially stronger signal than others. We quantify this via the ratio Var(RV)/Mean(RV):
>
> **Table R6: Understanding when RV-filtering helps**
> | Setting | Filter Δ | P | G | RV Mean | RV Std | RV Var | RV Min | RV Max | Var/Mean |
> |---------|---------|---|---|---------|--------|--------|--------|--------|----------|
> | Sokoban, 14B | **+4.6** | 8 | 8 | 2.24 | 2.88 | 8.32 | 0.10 | 6.00 | **3.71** |
> | Sokoban, 3B | **+3.19** | 32 | 4 | 2.49 | 2.89 | 8.35 | 0.05 | 6.52 | **3.35** |
> | FrozenLake, 3B, GRPO | **−5.0** | 32 | 8 | 0.54 | 0.18 | 0.03 | 0.22 | 0.76 | **0.06** |
>
> When Var(RV) ≫ Mean(RV), the RV distribution is bimodal (high-signal vs. noise-only prompts) and filtering cleanly separates them. When Var(RV) ≪ Mean(RV), all prompts carry similar RV and filtering discards data at random — explaining FrozenLake GRPO (Δ = −5.0), where stochastic transitions flatten the cross-prompt RV structure. This ratio is computable from a single rollout batch, providing a cheap pre-training diagnostic.
>
> The analysis above shows our framework has **predictive power**: the Var(RV)/Mean(RV) ratio, computable from a single rollout batch, determines whether RV-filtering is appropriate — moving it from a heuristic to a **principled tool with a checkable applicability condition**.
>
> Given these additional experiments and clarifications, we hope the reviewer will consider maintaining or updating their assessment.

---

> > ### Author Rebuttal · Reviewer_EXYe · 2026-04-01
> >
> > I thank the authors for their detailed rebuttal. The response has properly addressed my concerns, and I have decided to maintain my score.

---

### Official Review · Reviewer_931D · 2026-03-13

**Soundness:** 3
**Presentation:** 4
**Significance:** 3
**Originality:** 3
**Overall Recommendation:** 4
**Confidence:** 3

**Summary:**

The paper focuses on the phenomenon of Reasoning Collapse in closed-loop, multi-turn LLM Agent RL. The paper proposes an information-theoretic framework to distinguish the reasoning collapse and template collapse. Based on this framework, they propose a novel Mutual Information (MI)-based retrieval diagnostic method to quantify the extent of collapse. Furthermore, the paper provides a theoretical explanation, and empirical results confirm the effectiveness of this approach across various RL algorithms, model scales, and multimodal settings.

**Compliance With Llm Reviewing Policy:**

Affirmed.

**Final Justification:**

The rebuttal properly addresses my concerns. I maintain my score.

**Key Questions For Authors:**

1. To estimate $Var(R|X)$, the method needs sampling $G$ trajectories per prompt, for example $G \geq 8$. While the paper argues that this aligns with the standard sampling logic of GRPO, maintaining such a high sampling frequency across training runs poses a burden on both VRAM footprint and running time. The authors should provide more details on whether the performance gains justify this increased computational expenditure.

**Limitations:**

yes

**Strengths And Weaknesses:**

Strengths:
1. The paper proposes a novel definition of reasoning collapse. It distinguishes the reasoning collapse and template collapse from an information-theoretic perspective.
2. The experiment is extensive. The experiments cover PPO/GRPO, model sizes ranging from 0.5B to 7B, and even include multi-model agents. It demonstrates both the universality of this phenomenon and the effectiveness of the method.

Weakness:
1. See the questions.

---

> ### Author Rebuttal · Authors · 2026-03-29
>
> We thank Reviewer 931D for the detailed feedback and for recognizing that our work proposes **a novel definition of reasoning collapse** with **extensive** experiments covering PPO/GRPO, model sizes ranging from 0.5B to 7B, and even multi-modal agents.
>
> ---
>
> ### [W1 / Q1] Computational Cost of Sampling G≥16 Trajectories Per Prompt
>
> We appreciate this practical question. We clarify two points and provide new experimental evidence.
>
> **The total rollout budget is fixed, not increased.** Our training protocol generates a constant 128 trajectories per iteration. The P×G (prompt v.s. generated trajectories per prompt) split is a *partitioning* of this fixed budget:
>
> | Config | Prompts (P) | Traj/Prompt (G) | Total Traj | Rollout Cost |
> |--------|------------|----------------|-----------|-------------|
> | 128×1  | 128        | 1              | 128       | 1.0×        |
> | 64×2   | 64         | 2              | 128       | 1.0×        |
> | 32×4   | 32         | 4              | 128       | 1.0×        |
> | 16×8   | 16         | 8              | 128       | 1.0×        |
> | 8×16   | 8          | 16             | 128       | 1.0×        |
>
> Since all configurations generate the same number of trajectories, rollout number keeps identical, which dominates wall-clock time and VRAM similar.
>
> **RV-filtering further reduces update cost.** With keep rate TopP < 1, fewer trajectory groups enter the gradient computation. The filtering step itself (computing per-prompt reward variance and sorting) takes <~0.1% of iteration time.
>
> **New experiment: P×G sweep with and without filtering.** We ran the full sweep on Sokoban (Qwen2.5-3B):
>
> **Table R1: P×G Sweep on Sokoban (Qwen2.5-3B). NF means non-filtering, F means RV-Filtering (p=0.9).**
>
> | Config | Perf. (%) (NF / F / Δ) | Avg Step Time (s) (NF / F / Δ%) | VRAM (GB) (NF / F / Δ) |
> |--------|---------------------|------------------------|------------------------|
> | 128×1  | 23.63 / N/A / —     | 89.82 / N/A / —        | 201.80 / N/A / —       |
> | 64×2   | 18.75 / 27.34 / +8.59 | 91.84 / 64.93 / −29.3% | 201.39 / 201.83 / +0.44 |
> | 32×4   | 24.22 / 27.41 / +3.19 | 89.78 / 52.63 / −41.4% | 202.11 / 201.67 / −0.44 |
> | 8×16   | 15.62 / 23.63 / +8.01 | 89.23 / 65.87 / −26.2% | 201.54 / 201.90 / +0.36 |
>
> Three observations: (i) wall-clock time is constant across configurations for non-fitering experiments; (ii) RV-filtering consistently reduces per-step time by 26–41%; (iii) configurations with G≥4 and RV-filtering match or outperform the 128×1 baseline. The performance gains come at **no additional cost**, and filtering even provides a net compute saving.
>
> We have added this analysis to the revised paper (Section 4). We further analyze when filtering helps vs. hurts in our responses to NKT7 and EXYe (cross-prompt RV heterogeneity as predictor).
>
> Given these results and clarifications, we hope the reviewer will consider updating their assessment.

---

> > ### Author Rebuttal · Reviewer_931D · 2026-03-31
> >
> > The rebuttal properly addresses my concerns. I maintain my score.

---

### Decision · Program_Chairs · 2026-04-30

**Decision:**

Accept (spotlight)

**Comment:**

All reviewers found this a deep and insightful contribution to the area of reasoning models. It identifies a mechanism by which reasoning algorithm collapse without it being trackable by straightforward measures like entropy. The authors propose a principled diagnostic method, and a mechanistic explanation that all reviewers found convincing.  The experiments were persuasive too, and overall no major weaknesses seem present.